# Structural basis of ClC-3 transporter inhibition by TMEM9 and PtdIns(3,5)P$_2$

**Marina Schrecker**[1,9], **Yeeun Son** [1,2,9], **Rosa Planells-Cases** [3], **Sumanta Kar** [3], **Viktoriia Vorobeva** [3,4], **Uwe Schulte** [5,6], **Bernd Fakler** [5,7], **Thomas J. Jentsch** [3,8] ✉ & **Richard K. Hite** [1] ✉

The trafficking and activity of endosomes relies on the exchange of chloride ions and protons by members of the CLC family of chloride channels and transporters; mutations of the genes encoding these transporters are associated with numerous diseases. Despite their critical roles, the mechanisms by which CLC transporters are regulated are poorly understood. Here we show that two related accessory β-subunits, TMEM9 and TMEM9B, directly interact with ClC-3, ClC-4 and ClC-5. Cryo-electron microscopy structures reveal that TMEM9 inhibits ClC-3 by sealing the cytosolic entrance to the Cl$^-$ ion pathway. Unexpectedly, we find that phosphatidylinositol 3,5-bisphosphate (PtdIns(3,5)P$_2$) stabilizes the interaction between TMEM9 and ClC-3 and is required for proper regulation of ClC-3 by TMEM9. Collectively, our findings reveal that TMEM9 and PtdIns(3,5)P$_2$ collaborate to regulate endosomal ion homeostasis by modulating the activity of ClC-3.

The acidic organelles of the endolysosomal system require precise maintenance of their luminal ion concentrations, including protons, Cl$^-$ and Ca$^{2+}$, to ensure proper endosomal trafficking and function[1-4]. Consequently, dysregulation of endolysosomal ion homeostasis is associated with numerous pathologies, ranging from kidney stones to neurodegeneration[5-9]. Five members of the CLC family of Cl$^-$ channels and Cl$^-$/H$^+$ exchangers, ClC-3, ClC-4, ClC-5, ClC-6 and ClC-7, have critical roles in endosomes and lysosomes, catalyzing the exchange of two Cl$^-$ ions for one H$^+$ ion[2,10-12]. CLC transporters are functional dimers, with each protomer possessing an independent Cl$^-$ ion pathway[2,13,14]. Located within each Cl$^-$ ion pathway is a conserved glutamate residue, known as the gating glutamate, that is essential for coupled Cl$^-$ and H$^+$ transport[13,15-18]. Structures of CLC transporters have revealed that the gating glutamate adopts several conformations that correspond to distinct states in the transport cycle[2,13,14,16,19]. However, despite numerous structural investigations and their prominent roles in endolysosomal ion homeostasis, the mechanisms by which CLC transporters are regulated are poorly understood.

Many transport proteins are regulated by accessory β-subunits that can participate in membrane trafficking and/or directly modulate transport. CLCs have three known β-subunits: OSTM1, an obligatory β-subunit for the lysosomal ClC-7 transporter, barttin, an obligatory β-subunit for the plasma membrane channels ClC-Ka and ClC-Kb, and glial CAM, a facultative β-subunit for ClC-2 (refs. 20–28). How OSTM1 and barttin regulate the activities of the CLCs remains incompletely understood. Whether other β-subunits contribute to the trafficking and regulation of other CLCs is also unknown.

We recently found that the ClC-3, ClC-4 and ClC-5 clade of endosomal CLC transporters requires the accessory β-subunit TMEM9, which we refer to as T9A or its closely related homolog TMEM9B (T9B) for proper activity in cells and animals[29]. Consistent with a recent report that T9B suppresses plasma membrane currents of ClC-3 and ClC-4 (ref. 30), we found that T9A and T9B strongly reduce the plasma membrane expression of members of the ClC-3, ClC-4 and ClC-5 clade[29]. Moreover, we found that T9A and T9B directly inhibit CLC ion transport through a mechanism that requires their C-terminal domains (CTDs)[29].

[1]Structural Biology Program, Memorial Sloan Kettering Cancer Center, New York, NY, USA. [2]BCMB Allied Program, Weill Cornell Graduate School, New York, NY, USA. [3]Leibniz-Forschungsinstitut für Molekulare Pharmakologie (FMP), Berlin, Germany. [4]Graduate program of the Free University, Berlin, Germany. [5]Institute of Physiology, Faculty of Medicine, University of Freiburg, Freiburg, Germany. [6]Logopharm GmbH, March-Buchheim, Germany. [7]Signaling Research Centers BIOSS and CIBSS, Freiburg, Germany. [8]Neurocure Cluster of Excellence, Charité Universitätsmedizin, Berlin, Germany. [9]These authors contributed equally: Marina Schrecker, Yeeun Son. ✉e-mail: Jentsch@fmp-berlin.de; hiter@mskcc.org

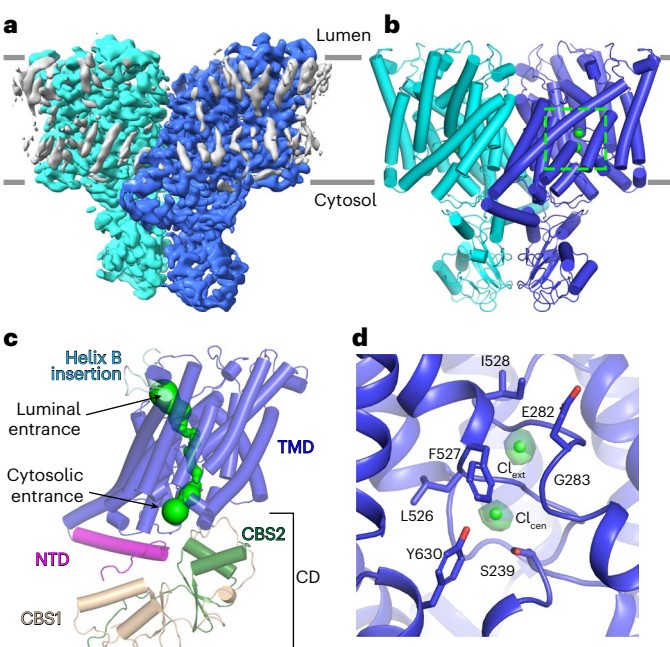

**Fig. 1 | Structure of human ClC-3. a,b**, Cryo-EM density map (**a**) and fitted atomic model (**b**) of ClC-3 colored by protomer. Cl⁻ ions are shown as green spheres. The dashed box in **b** corresponds to **d. c**, ClC-3 protomer with the NTD colored in magenta, the TMD in blue, the helix B insertion in light blue, CBS1 in tan and CBS2 in dark green. The Cl⁻ ion pathway is shown as a green surface. **d**, Cl⁻-binding sites in the Cl⁻ ion pathway of one protomer. Densities corresponding to Cl⁻ ions are shown as green isosurfaces and contoured at $4.0\sigma$.

To resolve the mechanism by which T9A and T9B regulate the activity of ClC-3, ClC-4 and ClC-5, we determined structures of ClC-3 alone and in complex with T9A, finding that T9A occludes the Cl⁻ ion pathway and that inhibition requires the endolysosomal signaling lipid phosphatidylinositol 3,5-bisphosphate (PtdIns(3,5)P₂).

## Results

### Structure of human ClC-3

To determine how T9A and T9B regulate ClC-3, ClC-4 and ClC-5, we collected cryo-electron microscopy (cryo-EM) images of ligand-free human ClC-3 alone and in complex with human T9A. We focused on ClC-3 because it forms stable complexes with T9A and T9B, whereas detergent-solubilized ClC-4 and ClC-5 only partially associate with T9A and T9B (Extended Data Fig. 1). Analysis of the images of ClC-3 alone yielded a $C_2$ symmetric reconstruction of dimeric ClC-3 at a resolution of 2.5 Å (Fig. 1a,b, Extended Data Fig. 2 and Table 1). Each ClC-3 protomer comprises a cytosolic N-terminal domain (NTD), a transmembrane domain (TMD) and a cytosolic domain (CD) (Fig. 1c). The NTD contains helix A and a loop that is sandwiched between the TMD and the CD. The ClC-3 TMD adopts the canonical CLC fold with a Cl⁻ ion pathway passing through each protomer. In contrast to the continuous helical conformation of helix B in ClC-K, ClC-2, ClC-6 or ClC-7 (refs. 27,31–33), helix B of ClC-3 is separated into two segments by a 32-residue insertion that we call the helix B insertion (Extended Data Fig. 3). The helix B insertion adopts a partially ordered conformation that is stabilized by two disulfide bonds. The ClC-3 CD contains two cystathionine-β-synthase (CBS) domains that establish an adenosine-triphosphate-binding site[34,35], in which no density corresponding to a bound adenine nucleotide was observed, indicating that we resolved a ligand-free structure (Fig. 1c).

The luminal and cytosolic entrances to the Cl⁻ ion pathway are solvent accessible. We assigned density peaks occupying the central and external Cl⁻-binding sites as Cl⁻ ions (Fig. 1d). The central Cl⁻-binding site is formed by the side chains of S239 and Y630 and the backbone nitrogen of L526 and the external binding site is formed by backbone nitrogen atoms from E282, G283, F527 and I528. Nonprotein densities present near the internal Cl⁻-binding site were modeled as water molecules as their identities were unclear. The conserved gating glutamate, E282, adopts an outward conformation, establishing a constriction with a minimum radius of 0.4 Å between the external Cl⁻-binding site and the luminal entrance to the Cl⁻ ion pathway. Overall, these features are consistent with reported structures of other CLC transporters, including recently reported structures of mouse ClC-3 (ref. 35), indicating that this structure of human ClC-3 reflects a transport-competent state.

### Structure of ClC-3 in complex with T9A

Analysis of cryo-EM images of ClC-3 in complex with T9A revealed four classes that can be distinguished by the densities corresponding to T9A (Fig. 2a, Extended Data Figs. 4 and 5 and Table 1). We identified a two-fold symmetric class in which the luminal domain (LD), TMD and CD of both T9A protomers are resolved, which we call ClC-3/T9A (Fig. 2a and Extended Data Fig. 5a,e). We also identified two asymmetric classes in which the densities corresponding to the T9A protomers were incompletely resolved, with the LD and/or CD of T9A being partially or completely disordered (Extended Data Fig. 5c,d,g,h). The weak density for these domains suggests that they are more dynamic than the TMD. Despite being coexpressed with T9A, we did not observe any densities corresponding to the T9A protomers in the fourth class, possibly because of T9A dissociation during purification (Extended Data Fig. 5b,f). To distinguish this fourth class from the structure of ClC-3 determined in the absence of T9A, we call this class ClC-3/noT9A. Because of its completeness, we focus our discussion primarily on ClC-3/T9A, which achieved a resolution of 2.9 Å with $C_2$ symmetry imposed (Table 1).

In ClC-3/T9A, T9A wraps around ClC-3, interacting with the luminal, transmembrane and cytosolic faces of the ClC-3 TMD. Notably, we do not observe interactions between the two T9A protomers. The T9A LD comprises a three-stranded β-sheet and two short α-helices that are stabilized by four disulfide bonds (Fig. 2b). The residues that establish these disulfide bonds are conserved in human T9B, suggesting that the fold of the LD is conserved (Fig. 2c). The T9A LD interacts primarily with the ClC-3 helix B insertion, forming both polar and nonpolar interactions, including the embedding of F181 of ClC-3 into a hydrophobic groove on T9A and an extended π-stacking interaction between R28 and R83 of T9A with W174 of ClC-3 (Fig. 2a,d). The interactions with T9A stabilize the ClC-3 helix B insertion in a more ordered state in ClC-3/T9A, allowing us to model the entire domain. The sequence of the helix B insertion is highly conserved across ClC-3, ClC-4 and ClC-5 but not ClC-6 or ClC-7, suggesting that the helix B insertion is a characteristic feature among CLC transporters that interact with T9A and T9B (Extended Data Fig. 3).

The single membrane-spanning helix of T9A, helix H3, extends along the periphery of the ClC-3 TMD, forming interactions with helices B, C, E and I of ClC-3 (Fig. 2e). Despite being only a single helix, its interaction with ClC-3 buries a large surface area because of its highly tilted orientation in the membrane. The interactions between helix H3 and the TMD of ClC-3 are almost exclusively hydrophobic, with residues such as I92 and Y95 of T9A occupying hydrophobic grooves on the surface of ClC-3.

The T9A CD consists of the H4 helix that extends along the cytosolic face of the ClC-3 TMD and four residues at the extreme C terminus, which we define as the CTD (Fig. 3a). The T9A CD is connected to the membrane-spanning H3 helix by an extended linker that is disordered in the structure and contains a stretch of negatively charged residues and several phosphorylation sites that contribute to the trafficking of ClC-3 and T9A[29]. The H4 helix interacts with the cytosolic ends of helices C, D, E, J and R, the D–E linker, and the J–K linker. Among the residues on

**Table 1 | Cryo-EM data collection and refinement statistics**

| | 1. ClC-3 (EMD-47070) (PDB 9DOO) | 2. ClC-3/noT9A (EMD-47066) (PDB 9DNW) | 3. ClC-3/T9A T9A protomers A and B: complete (EMD-47067) (PDB 9DNX) | 4. ClC-3/T9A T9A protomer A: no CD T9A protomer B: no LD, no CD (EMD-47068) (PDB 9DNY) | 5. ClC-3/T9A T9A protomer A: complete T9A protomer B: no LD, no CD (EMD-47069) (PDB 9DNZ) |
|---|---|---|---|---|---|
| **Data collection and processing** | | | | | |
| Detector | Gatan K3 | TFS Falcon4i | TFS Falcon4i | TFS Falcon4i | TFS Falcon4i |
| Magnification | ×29,000 | ×165,000 | ×165,000 | ×165,000 | ×165,000 |
| Voltage (kV) | 300 | 300 | 300 | 300 | 300 |
| Energy filter slit width (eV) | | 10 | 10 | 10 | 10 |
| Electron exposure (e⁻ per Å²) | 66 | 59.63 | 59.63 | 59.63 | 59.63 |
| Defocus range (µm) | −0.7 to −2 | −0.5 to −1.5 | −0.5 to −1.5 | −0.5 to −1.5 | −0.5 to −1.5 |
| Super-resolution pixel size (Å) | 0.413 | | | | |
| Final pixel size (Å) | 0.826 | 0.725 | 0.725 | 0.725 | 0.725 |
| Symmetry imposed | $C_2$ | $C_2$ | $C_2$ | $C_1$ | $C_1$ |
| Initial particle images (no.) | 1,446,105 | 10,137,040 | 10,137,040 | 10,137,040 | 10,137,040 |
| Final particle images (no.) | 2,19,662 | 1,48,161 | 94,011 | 91,755 | 71,754 |
| Map resolution (Å) | 2.54 | 2.9 | 2.86 | 3.01 | 3.16 |
| FSC threshold | 0.143 | 0.143 | 0.143 | 0.143 | 0.143 |
| **Refinement** | | | | | |
| Model resolution (Å) | | | | | |
| 0.5 FSC threshold | 2.51 | 2.79 | 2.84 | 2.96 | 3.13 |
| Map sharpening $B$ factor (Å²) | −30 | −30 | −30 | −30 | −30 |
| Model composition | | | | | |
| Nonhydrogen atoms | 11,130 | 11,208 | 13,280 | 12,326 | 12,524 |
| Protein residues | 1,394 | 1,399 | 1,656 | 1,540 | 1,562 |
| Ligands | 8 | 10 | 10 | 10 | 10 |
| Mean $B$ factors (Å²) | | | | | |
| Protein | 58 | 26.1 | 67.4 | 46 | 63.8 |
| Ligand | 52.1 | 15.5 | 52.3 | 35.2 | 53.3 |
| Water | 26.4 | | | | |
| Root-mean-square deviations | | | | | |
| Bond lengths (Å) | 0.002 | 0.002 | 0.003 | 0.002 | 0.002 |
| Bond angles (°) | 0.426 | 0.449 | 0.477 | 0.476 | 0.499 |
| **Validation** | | | | | |
| MolProbity score | 1.07 | 1.24 | 1.41 | 1.4 | 1.35 |
| Clashscore | 2.83 | 4.54 | 5.35 | 5.96 | 5.34 |
| Poor rotamers (%) | 0.09 | 1.03 | 1.43 | 1.24 | 1.22 |
| Ramachandran plot | | | | | |
| Favored (%) | 98.48 | 98.48 | 98.77 | 98.95 | 98.44 |
| Allowed (%) | 1.52 | 1.52 | 1.23 | 1.05 | 1.56 |
| Disallowed (%) | 0 | 0 | 0 | 0 | 0 |

PDB, Protein Data Bank; FSC, Fourier shell correlation.

the T9A H4 helix that interact with ClC-3 are W164, which contacts F232 and W260 of ClC-3, and F176, which is inserted into a groove involving K230, P234, P427 and K521 of ClC-3 (Fig. 3b). Additionally, the side chain of R172 of T9A interacts with the backbone oxygen atoms of K230 and V231 of ClC-3. Collectively, these interactions position the H4 helix such that the four-residue CTD can extend from helix H4 along the side of ClC-3 towards the CD, where S183 of T9A interacts with the backbone of F88 and the side chain of K798 of ClC-3 (Fig. 3c).

## T9A and OSTM1 bind to distinct interfaces
To further investigate the specificity of T9A and T9B for ClC-3, ClC-4 and ClC-5 and of OSTM1 for ClC-7, we compared ClC-3/T9A with a structure

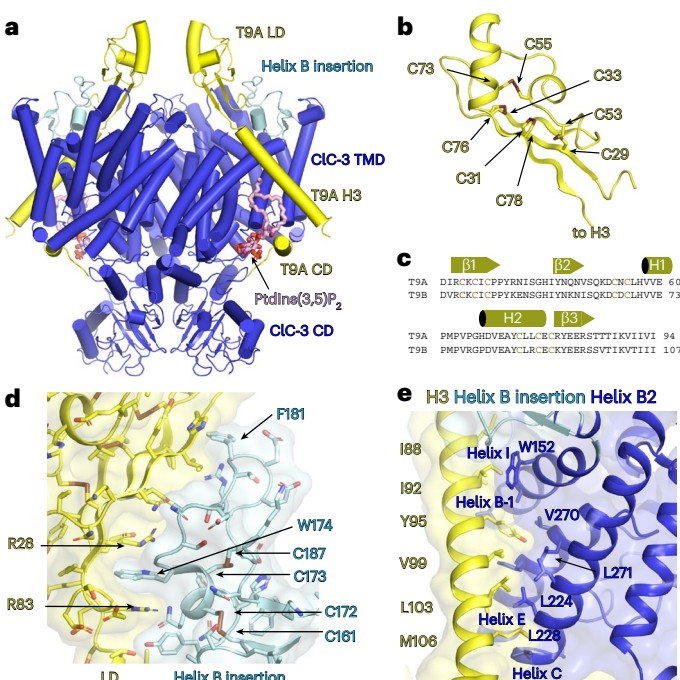

**Fig. 2 | Structure of human ClC-3 in complex with T9A. a**, ClC-3/T9A colored by subunit with T9A in yellow, ClC-3 in blue and the ClC-3 helix B insertion in light blue. PtdIns(3,5)P$_2$ is shown in pink. **b**, T9A LD with residues that form disulfide bonds shown as sticks. **c**, Sequence alignment of human T9A and T9B LDs. Secondary-structure elements and residues that form disulfide bonds in T9A are highlighted. **d**, Interactions between T9A LD and the helix B insertion of ClC-3. **e**, Interactions between H3 of T9A and the TMD of ClC-3.

of ClC-7/OSTM1 (ref. 27). Although T9A and OSTM1 are both single-pass transmembrane proteins with disulfide-stabilized LDs that serve as obligatory β-subunits[23,24,29], T9A and OSTM1 bind to distinct surfaces on their corresponding CLCs (Extended Data Fig. 6a–d). For example, whereas the T9A LD binds to the ClC-3 helix B insertion, the LDs of OSTM1 form few interactions with ClC-7 and instead form an extensive homodimeric interface that helps to stabilize ClC-7/OSTM1 in the harsh environment of the lysosome[27]. The TMDs of T9A and OSTM1 also bind in dissimilar manners; T9A traverses along the side of the ClC-3 TMD in a highly tilted manner, while OSTM1 binds in upright manner to the extreme periphery of the ClC-7 TMD. Lastly, in contrast to the extensive interactions formed between the T9A CD and ClC-3, no evidence for an interaction between the OSTM1 CD and ClC-7 has been observed in the reported structures[27,28]. Thus, despite their common origin, ClC-3, ClC-4 and ClC-5 have evolved unique features that enable interactions with T9A and T9B compared to those that have evolved in ClC-7 to facilitate its interactions with OSTM1.

### The T9A CD plugs the Cl⁻ ion pathway

We next sought to understand how T9A inhibits ClC-3 without altering the global conformation of ClC-3. Comparing the ClC-3 structures with and without T9A demonstrated that T9A does not change the conformation of the gating glutamate, E282, or the positions of the ions in the central and external Cl⁻-binding sites in ClC-3/T9A (Extended Data Fig. 7a,b). Rather, T9A inhibits ClC-3 by sealing the cytosolic entrance to the Cl⁻ ion pathway. Whereas both entrances of the Cl⁻ ion pathway are solvent accessible in the absence of T9A, the CDs of T9A constrict the cytosolic entrances to the Cl⁻ pathway in ClC-3/T9A to a minimum radius of 0.9 Å, which is too narrow to accommodate a Cl⁻ ion (Fig. 3d–f). We, therefore, assign ClC-3/T9A as an inhibited state, with the T9A CD forming the gate. In addition to ClC-3/T9A, we observed two classes of the complex in which one or both T9A CDs are disordered (Extended

Data Fig. 7c,d). The TMDs are resolved in these classes, indicating that the disordered CDs are present but not bound to ClC-3, which is consistent with immunoprecipitation analysis that demonstrated that the LD and the TMD but not the CD are required for the interaction between ClC-3 and T9A[29]. The disengagement of the T9A CD exposes the cytosolic entrance to the Cl⁻ ion pathway, indicating that these correspond to disinhibited states. We, therefore, propose that the T9A CD blocks access to the Cl⁻ ion pathway through a ball-and-chain-like mechanism.

To probe the role of the T9A CD in the regulation of ClC-3, we examined the effects of genetic perturbations to the interface between ClC-3 and the T9A CD on ClC-3 activity. We used the appearance of enlarged vacuoles as a readout of ClC-3 activity in endosomes, as expression of the endosome-targeted ClC-3a isoform leads to enlarged endosomes[29,36,37], a phenotype that can be suppressed by coexpression with T9A or T9B (ref. 29). We first generated alanine substitutions of residues W164 and R172 of T9A, two residues in the T9A CD that form extensive interactions with ClC-3 (Fig. 4a,b). Whereas vacuolization induced by ClC-3 expression was suppressed in cells coexpressing wild-type (WT) T9A, enlarged vacuoles were present in cells coexpressing the T9A$^{W164A}$ or T9A$^{R172A}$ mutants (Fig. 4c–e). These results are consistent with the model that W164 and R172 are necessary for the inhibitory effect of T9A on ClC-3. We next asked whether W260 of ClC-3, which forms a π–π interaction with W164 of T9A, was likewise required for T9A to inhibit ClC-3 (Fig. 4a). Indeed, substituting W260 of ClC-3 with alanine prevented T9A from suppressing vacuolization (Fig. 4e). Collectively, these data highlight the critical role of the T9A CD in inhibiting ClC-3. Moreover, these results are consistent with scanning mutagenesis experiments that identified several stretches of residues in the T9B CD that abrogated the ability of T9B to inhibit ClC-3 (ref. 29). Together, these findings suggest that T9A and T9B inhibit ClC-3 through a conserved mechanism.

### PtdIns(3,5)P$_2$ is required for the inhibition of ClC-3 by T9A

A nonprotein density is present in the ClC-3/T9A density map near the interface between ClC-3 and the T9A helix H4. We modeled this density as a copurified PtdIns(3,5)P$_2$, a lipid synthesized from phosphatidylinositol 3-phosphate ((PtdIns3P) by PIKfyve in endosomes and lysosomes[38] (Figs. 2a and 5a,b and Extended Data Figs. 5 and 6g,h). Densities corresponding to PtdIns(3,5)P$_2$ are also resolved in the asymmetric ClC-3/T9A reconstructions (Extended Data Fig. 5). A weaker density is also resolved in the same position in ClC-3/noT9A, suggesting that PtdIns(3,5)P$_2$ may bind with lower occupancy when T9A is not stably bound. In accordance with T9A influencing the binding of PtdIns(3,5)P$_2$, no densities are present in the PtdIns(3,5)P$_2$-binding site in the ClC-3-only reconstruction or in reconstructions of mouse ClC-3 (ref. 35).

In ClC-3/T9A, PtdIns(3,5)P$_2$ interacts with both ClC-3 and the T9A CD. The phosphate groups at the 1 and 3 positions of the PtdIns(3,5)P$_2$ inositol sugar bind to ClC-3, whereas the phosphate at the 5 position interacts with the T9A CD (Fig. 5b). The phosphate at the 1 position is coordinated by the backbone nitrogen atoms of K259 and W260 of ClC-3 and the side chain of K259 of ClC-3. The phosphate at the 3 position binds to the backbone nitrogen of G255 of ClC-3 and the side chains of R254, N294, Y298 and K310 of ClC-3. The phosphate at the 5 position interacts with the side chains of R163 and Q167 from the T9A H4 helix. Thus, PtdIns(3,5)P$_2$ appears to serve as a bridge between ClC-3 and the T9A CD, raising the possibility that PtdIns(3,5)P$_2$ influences the equilibrium between the bound and unbound states of the T9A CD to regulate ClC-3 activity.

Using the Venus–ClC-3a/T9A vacuolization assay described above, we investigated the role of PtdIns(3,5)P$_2$ in the T9A-mediated inhibition of endosomal ClC-3. To ensure specificity for ClC-3, experiments were performed in HeLa cells lacking ClC-7, a lysosomal Cl⁻/H⁺ exchanger that can generate enlarged endosomes when PtdIns(3,5)P$_2$ is depleted[39]. Expression of Venus–ClC-3a led to the generation of enlarged vacuoles in these cells (Fig. 5c)[29]. Whereas coexpression of

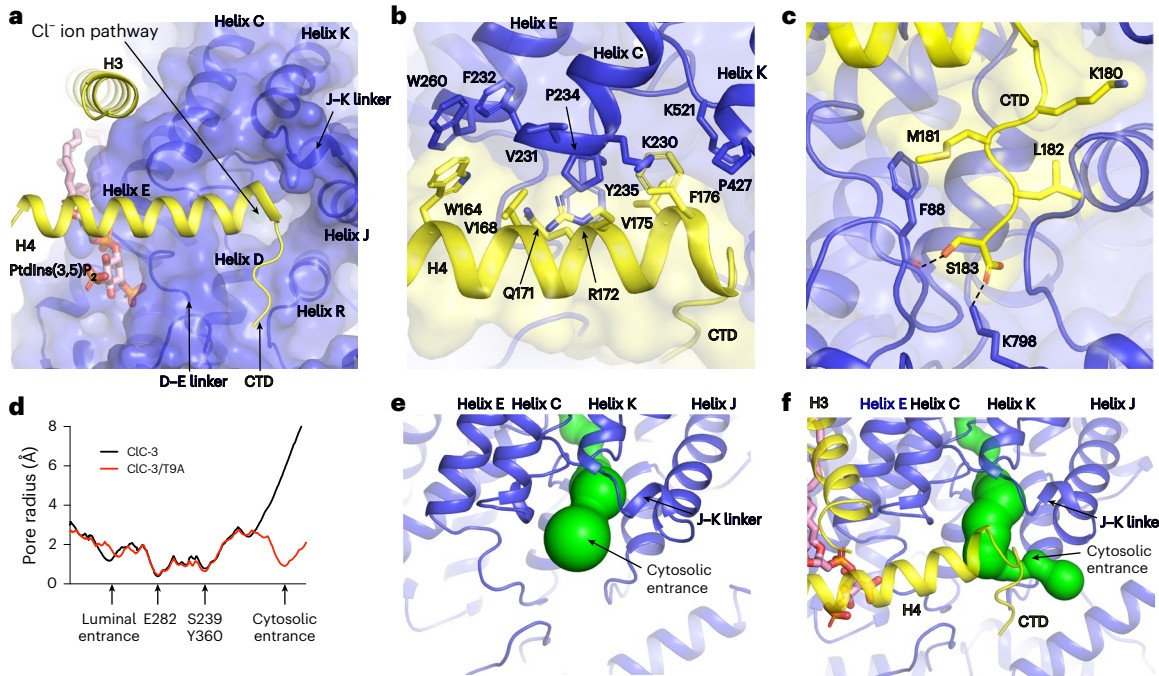

**Fig. 3 | The T9A CD plugs the Cl⁻ ion pathway. a**, The T9A CD, consisting of helix H4 and the four-residue CTD, binds to ClC-3 near the cytosolic entrance to the Cl⁻ ion pathway. **b**, Interactions between T9A helix H4 and ClC-3. **c**, Interactions between the T9A CTD and ClC-3. **d**, Radii of Cl⁻ ion pathways of ClC-3 (black) and ClC-3/T9A (red). **e,f**, The cytosolic entrance of the Cl⁻ ion pathway is open in ClC-3 (**e**) and sealed in ClC-3/T9A (**f**). Cl⁻ ion pathways are shown as green surfaces.

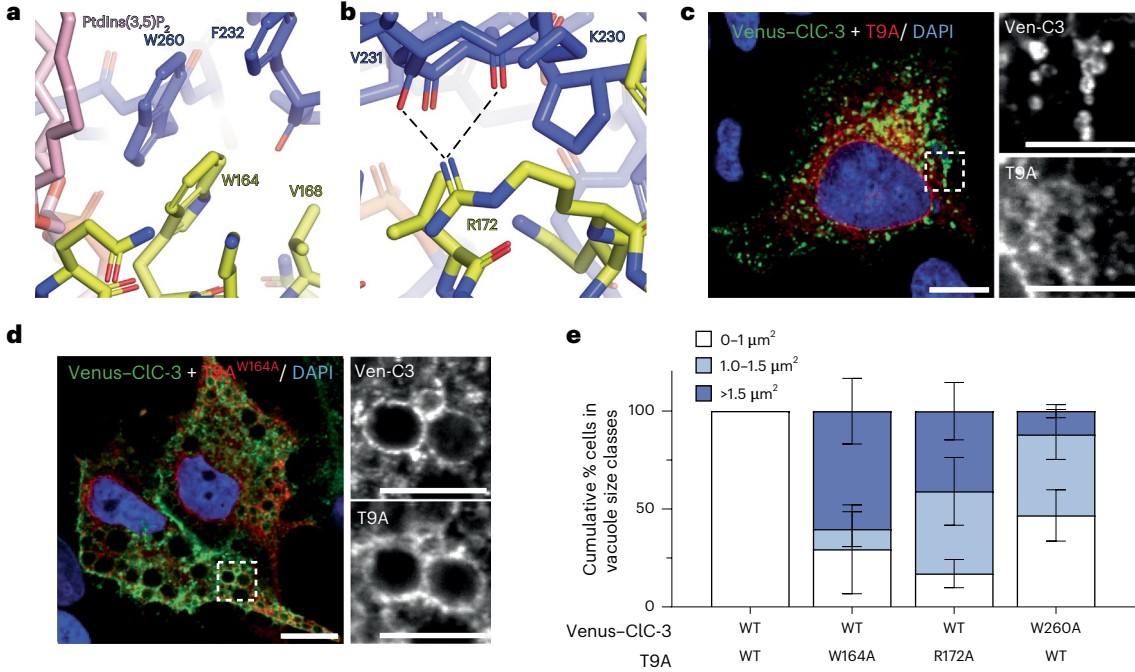

**Fig. 4 | The T9A CD is necessary for inhibiting ClC-3. a**, p–p interaction between W164 of T9A and W260 of ClC-3. **b**, R172 of T9A interacts with the backbone oxygen atoms of K230 and V231 of ClC-3. Hydrogen bonds are shown as dashed lines. **c,d**, Representative confocal images of ClC-3a-dependent endosomal vacuolization in cells coexpressing Venus–ClC-3a and T9A (**c**) or Venus–ClC-3a and T9A^W164A (**d**). Venus–ClC-3a, green; T9A, red; DAPI, blue. Right: zoomed-in view of boxed regions for Venus–ClC-3a (Ven-C3; top) and T9A (bottom). Scale bars, 10 μm and 5 μm (enlarged areas). **e**, Cumulative percentage of cells displaying punctae (<1 μm²), ≥5 moderate size vacuoles (1–1.5 μm²) per cell or ≥5 large vacuoles (>1.5 μm²) per cell. Vacuolization was investigated in fixed immunostained cells coexpressing Venus–ClC-3a with T9A, Venus–ClC-3a with T9A^W164A, Venus–ClC-3a with T9A^R172A and Venus–ClC-3a^W260A with T9A. Data are shown as the mean ± s.d. from three independent experiments (biological replicates).

T9A suppressed this vacuolization, enlarged vacuoles were observed in cells treated with apilimod or YM-201636, two structurally distinct inhibitors of PtdIns(3,5)P₂ generation by PIKfyve (Fig. 5d,e,i and Extended Data Fig. 8a,b). Notably, apilimod and YM-201636 had no impact on vacuolization in cells coexpressing T9A and the transport-deficient ClC-3^td (E339A) mutant, which, like similar mutants in other CLCs, almost completely abolishes ion transport[24,35] (Fig. 5f,i and Extended Data Fig. 8c,d). Hence, depletion of PtdIns(3,5)P₂ by

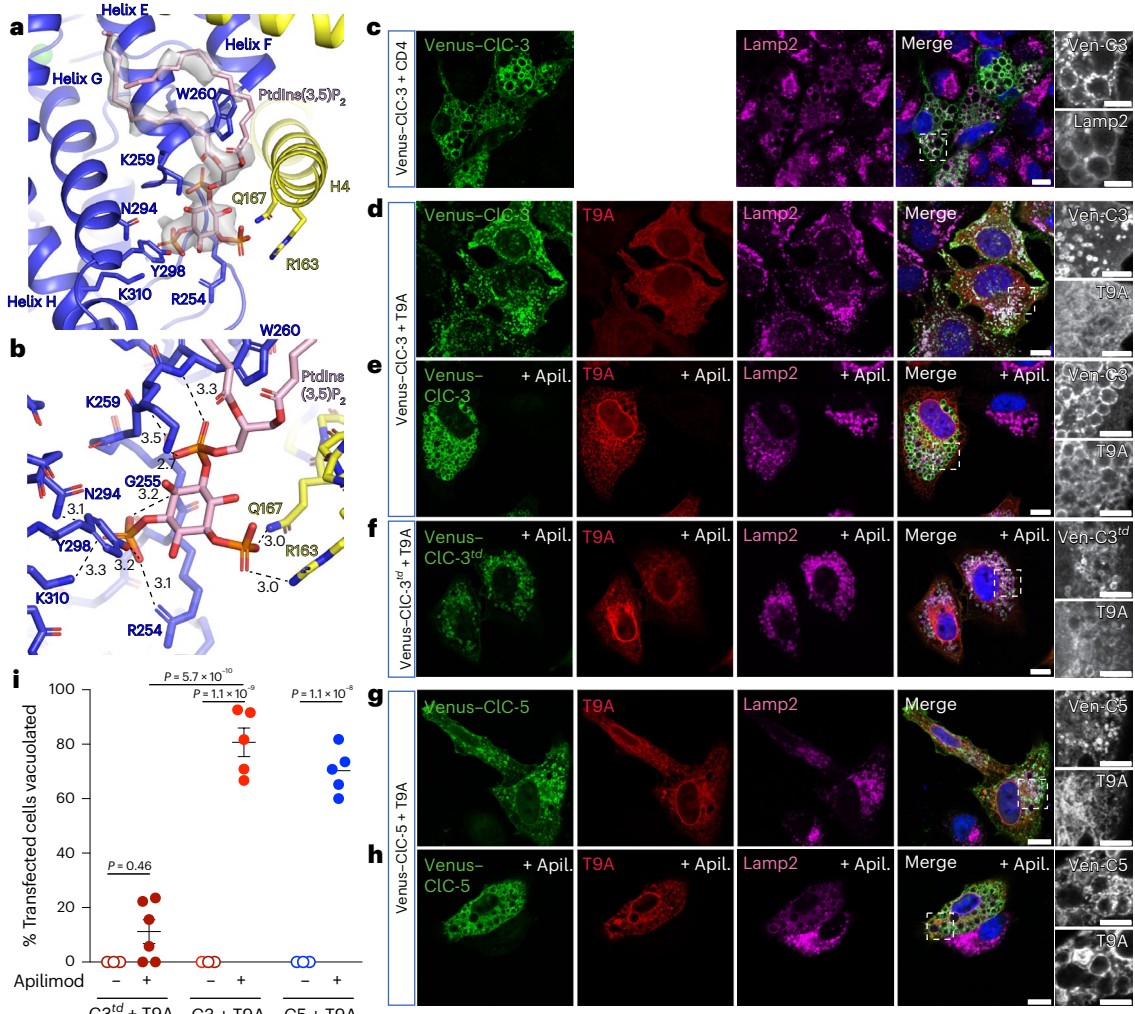

**Fig. 5 | Inhibition of ClC-3 by T9A requires PtdIns(3,5)P₂. a,** Cryo-EM density for bound PtdIns(3,5)P₂ displayed as gray isosurface and contoured at 2.5σ. **b,** Coordination of bound PtdIns(3,5)P₂ by ClC-3 and T9A. Dashed lines correspond to hydrogen bonds. **c–f,** Apilimod elicits endosomal vacuolization by weakening inhibition of ClC-3 by T9A. Representative confocal images of Venus–ClC-3a (green), T9A (red) and Lamp2 (magenta) from ClC-7-KO HeLa cells coexpressing Venus–ClC-3a and CD4 (**c**; control), Venus–ClC-3a and T9A (**d**), Venus–ClC-3a and T9A treated with 100 nM apilimod for 4 h (**e**) or Venus–ClC-3a[td] and T9A treated with 100 nM apilimod for 4 h (**f**). Right: merged images. Right: zoomed-in views of boxed regions for Venus–ClC-3a (Ven-C3; top) and Lamp2 (**c**) or T9A (**d–f**) (bottom). Scale bars, 10 μm and 5 μm (enlarged areas). **g,h,** Apilimod elicits vacuolization by relieving T9A-mediated block of

endosomally targeted ClC-5/T9A. Representative confocal images of Venus–ClC-5 (green), T9A (red) and Lamp2 (magenta) from ClC-7-KO HeLa cells coexpressing Venus–ClC-5 and T9A (**g**) or Venus–ClC-5 and T9A treated with 100 nM apilimod for 4 h (**h**). Right: merged image. Right: zoomed-in views of boxed regions for Venus–ClC-5 (Ven-C5; top) and T9A (**g,h**; bottom). Scale bars, 10 μm, and 5 μm (enlarged areas). Apil., apilimod. **i,** Fraction of cells displaying ≥5 large vacuoles (>1.5 μm²) per cell after 4-h treatment with 100 nM apilimod (or DMSO as vehicle control) for cells coexpressing T9A with Venus–ClC-3a (DMSO, n = 3; apilimod, n = 5), Venus–ClC-3a[td] (DMSO, n = 3; apilimod, n = 6) or Venus–ClC-5 (DMSO, n = 3; apilimod, n = 5). Each data point represents an independent experiment (biological replicate). Data are shown as the mean ± s.e.m. Statistical analysis was conducted using a two-way analysis of variance with multiple comparisons.

apilimod or YM-201636 elicits vacuolization by increasing ClC-3/T9A ion transport activity.

T9 proteins inhibit the three members of the ClC-3, ClC-4 and ClC-5 clade of endosomal chloride–proton exchangers[29]. ClC-3, ClC-4 and ClC-5 are highly homologous and the residues in ClC-3 that coordinate PtdIns(3,5)P₂ are conserved in ClC-4 and ClC-5. To determine whether PtdIns(3,5)P₂ has a conserved role in T9A-mediated inhibition across the ClC-3, ClC-4 and ClC-5 clade, we assessed the effect of apilimod on the inhibition of ClC-5 by T9A. Unlike ClC-3, trafficking of ClC-5 to endosomes requires coexpression of T9A or T9B; thus, expression of ClC-5 alone does not induce vacuolization[29]. However, coexpression of ClC-5 and T9A can yield enlarged vacuoles when mutations disrupt the association between ClC-5 and T9A[29], indicating that vacuolization can be used to monitor disinhibition of ClC-5. Similar to ClC-3, we observed enlarged vacuoles in ClC-7-deficient HeLa cells coexpressing

ClC-5 and T9A treated with apilimod but not in the untreated cells, demonstrating that PtdIns(3,5)P₂ has a conserved role in regulating the activity of the ClC-3, ClC-4 and ClC-5 clade (Fig. 5g–i).

We next explored how PtdIns(3,5)P₂ interacts with ClC-3 and T9A to stabilize the inhibited state. Alanine substitutions of R254 of ClC-3 or K310 of ClC-3, which participate in the coordination of the phosphate at the 3 position of PtdIns(3,5)P₂, diminished the inhibitory effect of T9A on ClC-3 (Extended Data Fig. 8e,f). R163 and Q167 of the T9A CD interact with the phosphate at the 5 position of PtdIns(3,5)P₂. Although substitution of R163 with alanine did not alter its ability to inhibit ClC-3, we identified a mutation of the gene encoding T9A that was observed in a person with lung cancer (National Cancer Institute Genomic Data Commons data portal), in which R163 is replaced with leucine, weakening the inhibitory effects of T9A on ClC-3 (Extended Data Fig. 8f). Thus, coordination of the phosphates at the 3 and 5

positions of the inositol sugar of PtdIns(3,5)$P_2$ by ClC-3 and T9A, respectively, is critical for the regulation of ClC-3.

Numerous mutations associated with human disease have been identified in *CLCN3*, which encodes ClC-3, and *CLCN4*, which encodes ClC-4 (refs. [6],[40]). Although some of the mutations affect ion transport by CLCs in isolation[6],[40], other mutations modulate the inhibitory effects of T9 proteins and can only be observed when they are coexpressed[29]. To gain insights into how disease-associated mutations in *CLCN3* and *CLCN4* perturb the inhibitory effects of T9 proteins, we mapped the mutations onto ClC-3/T9A. Although many of the mutations impacted residues near the interface with T9A, several were located near the PtdIns(3,5)$P_2$-binding site, highlighting the importance of PtdIns(3,5)$P_2$ in the regulation of CLCs (Extended Data Fig. 9). For example, the backbone oxygen atom of Y85 of ClC-3, which is substituted to cysteine in a neurological disease[6], interacts with the side chain of R254 of ClC-3 that binds the PtdIns(3,5)$P_2$ head group. D87 of ClC-3, which corresponds to D29 in ClC-4 that is substituted to glutamate in a neurological disease[40], and I252 of ClC-3, which was substituted to threonine in a person with neuropathy[6], also interact with R254 of ClC-3. Collectively, our results indicate that T9A and PtdIns(3,5)$P_2$ are dynamic regulators of endosomal CLC function and that disruption of the T9–CLC interface can lead to disease[29].

## Discussion

In this study, we reveal how the T9A CD regulates the activity of the ClC-3, ClC-4 and ClC-5 clade of endosomal chloride–proton exchangers through a mechanism that appears to be conserved in T9B. When bound to the cytosolic surface of ClC-3, the T9A CD physically occludes the cytosolic entrance to the Cl$^-$ ion pathway and inhibits ion transport. The T9A CD is flexibly tethered to the TMD, which is likely always associated with ClC-3 in endosomes. Notably, T9A engagement with ClC-3 is insufficient to fully inhibit its activity. Rather, the equilibrium between the bound, inhibited state and the unbound, disinhibited state of the T9A CD and, thus, the activity of ClC-3 are influenced by the endolysosomal signaling lipid, PtdIns(3,5)$P_2$. PtdIns(3,5)$P_2$ binds at the interface between ClC-3 and the T9A CD, stabilizing the bound, inhibited state. Regulation by PtdIns(3,5)$P_2$ abundance provides a mechanism to control CLC activity in endosomes through the generation and breakdown of PtdIns(3,5)$P_2$ by PIKfyve and FIG4, respectively[41],[42]. Post-translational modifications, such as phosphorylation of the T9A CD or interacting residues in CLCs[29], may further influence the equilibrium between bound and unbound states. These modifications would enable cells to finely tune endosomal CLC activity in a more specific way.

Recently, a structure of the related Cl$^-$ channel ClC-2 was resolved in which an inhibitory peptide from its N terminus was found to block the cytosolic entrance of the Cl$^-$ ion pathway (Extended Data Fig. 10)[32],[43],[44]. The binding site for the inhibitory N-terminal peptide of ClC-2 overlaps with the binding site for the T9A CD on ClC-3. Together with our work, these findings demonstrate that peptides from distinct origins can regulate CLCs by occluding the cytosolic entrance to the Cl$^-$ ion pathway in a manner similar to the ball-and-chain peptides of cation channels[45–47]. Future work may identify additional examples of interaction partners that regulate the activity of CLCs by binding to the cytosolic entrance to the Cl$^-$ ion pathway.

Phosphoinositol lipids, such as PtdIns(3,5)$P_2$, are regulators of diverse signaling pathways and can serve as direct modulators of numerous ion transport proteins. Structures of TRPML1 and TPC1 reveal that PtdIns(3,5)$P_2$ binding induces global conformational changes that gate the ion permeation pathways[48],[49]. Here, we establish an alternative paradigm where PtdIns(3,5)$P_2$ binding gates ion transport by acting as a molecular glue to stabilize the interaction between a transport protein and a regulatory domain of an accessory β-subunit. Similar to ClC-3, ClC-7 requires a β-subunit, OSTM1, for proper function and is inhibited by PtdIns(3,5)$P_2$ (ref. [39]). Suggestively, the PtdIns(3,5)$P_2$-binding site in ClC-3 partially overlaps with a site near the cytosolic end of helix E in ClC-7 that binds the related phosphatidylinositol, PtdIns3$P$[27] (Extended Data Fig. 6e,f). Future studies will be necessary to determine whether the overlap of the binding sites is coincidental or whether PtdIns(3,5)$P_2$ inhibits ClC-7 through a similar mechanism.

Collectively, our data highlight the role of T9 proteins and phosphatidylinositols in regulating endosomal CLC function. The mode of regulation enables a mechanistic understanding of pathological mutations in the genes encoding CLCs that lead to disease. In contrast to mutations that disrupt ion transport or alter the conformational landscape of ClC-3, we show that mutations that impact residues mediating interactions with accessory β-subunits and lipids can also lead to dysregulated CLC activity[29].

## Online content

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

## Methods

### Fluorescence-detection size-exclusion chromatography

Genes encoding human ClC-3 (P51790), human ClC-4 (P51793), human ClC-5 (P51795), human ClC-6 (P51797) and human ClC-7 (P51798) were synthesized by Twist Biosciences and subcloned into BacMam expression vectors with N-terminal mCerulean tags fused with a linker containing a PreScission protease site[50]. Genes encoding human T9A (Q9P0T7), human T9B (Q9NQ34) and human OSTM1 (Q86WC4) were synthesized by Twist Biosciences and subcloned into BacMam expression vectors with a C-terminal mVenus tag fused with a linker containing a PreScission protease site[50]. All construct sequences were validated by Sanger sequencing.

First, 800 µl of Expi293F cells with a cell density of $3 \times 10^6$ cells per ml were dispensed into a 96-deep-well plate (Corning, P-DW-20-C-S). Then, 0.8 µg of DNA and 2.4 µg of PEI 25K (Polysciences) were each mixed with 50 µl of Opti-MEM reduced-serum medium (Gibco) and incubated for 5 min at room temperature. For complex transfection, equal amounts of the plasmid encoding relevant N-terminal mCerulean-tagged CLC and the plasmid encoding C-terminal mVenus-tagged T9A or T9B were used. After incubation, DNA was combined with PEI 25K (Polysciences), incubated for 20 min at room temperature and then used for transfection. After 24-h incubation at 37 °C with continuous agitation, valproic acid sodium salt (Sigma-Aldrich, P4543) was added to a final concentration of 2.2 mM and cells were allowed to grow at 37 °C for an additional 24 h before harvesting. Cell pellets were washed in PBS and flash-frozen in liquid nitrogen. Expressed proteins were solubilized in 200 µl of buffer containing 2% LMNG (Anatrace, NG310), 20 mM HEPES pH 7.5 and 150 mM KCl supplemented with protease-inhibitor cocktail (1 mM PMSF, 2.5 mg ml$^{-1}$ aprotinin, 2.5 mg ml$^{-1}$ leupeptin and 1 mg ml$^{-1}$ pepstatin A) and DNase I. Solubilized proteins were separated by centrifugation at 21,130$g$ for 45 min. Separated proteins were injected to and monitored by fluorescence-detection size-exclusion chromatography (FSEC) on a Superose 6 Increase 10/300 GL (GE healthcare) in a buffer composed of 0.02% glyco-diosgenin (Anatrace, GDN101), 150 mM KCl, 20 mM HEPES pH 7.5 and 1 mM DTT (Goldbio, DTT10). mCerulean fluorescence was monitored at wavelengths of 433 and 475 nm for excitation and emission, respectively.

### Coimmunoprecipitation

First, 800 µl of Expi293F cells with a cell density of $3 \times 10^6$ cells per ml were dispensed into a 96-deep-well plate (Corning, P-DW-20-C-S). Then, 0.8 µg of DNA and 2.4 µg of PEI 25K (Polysciences) were each mixed with 50 µl of Opti-MEM reduced-serum medium (Gibco) and incubated for 5 min at room temperature. For complex transfection, equal amounts of the plasmid encoding relevant N-terminal mCerulean-tagged CLC and the plasmid encoding C-terminal twin-strep-tagged T9A was used. After incubation, DNA was combined with PEI 25K (Polysciences). incubated for 20 min at room temperature and then used for transfection. After 24-h incubation at 37 °C with continuous agitation, valproic acid sodium salt (Sigma-Aldrich, P4543) was added to a final concentration of 2.2 mM and cells were allowed to grow at 37 °C for an additional 24 h before harvesting. Cell pellets were washed in PBS and flash-frozen in liquid nitrogen. Expressed proteins were solubilized in 200 µl of buffer containing 2% LMNG (Anatrace, NG310), 20 mM HEPES pH 7.5 and 150 mM KCl supplemented with protease-inhibitor cocktail (1 mM PMSF, 2.5 mg ml$^{-1}$ aprotinin, 2.5 mg ml$^{-1}$ leupeptin and 1 mg ml$^{-1}$ pepstatin A) and DNase I. Solubilized proteins were separated by centrifugation at 21,130$g$ for 45 min, followed by binding to anti-GFP nanobody resin for 1.5 h at 4 °C, which was previously equilibrated with running buffer containing 0.02% LMNG, 20 mM HEPES pH 7.5, 150 mM KCl and 1 mM DTT. Anti-GFP nanobody affinity chromatography was performed by washing with 1 ml of running buffer. Bound proteins were eluted by incubating the anti-GFP nanobody resin with 1× Tris–glycine–SDS buffer, consisting of 25 mM Tris, 192 mM glycine and 0.1% SDS, at 95 °C for 10 min.

### Immunoblotting

Total protein was quantified using a BCA protein assay kit (Thermo Fisher Scientific, 23227) with the provided albumin standard used as a protein standard. Samples were resolved on 4–12% Bis–Tris gels (Thermo Fisher Scientific, NW04125BOX) and analyzed by standard immunoblotting techniques. Briefly, gels were transferred in transfer buffer (25 mM Tris base, 190 mM glycine and 20% methanol, pH 8.3) to nitrocellulose membranes (Bio-Rad, 1620115) and incubated with primary antibodies (strep-tag; Invitrogen, PA5-114453) for 3 h at room temperature. Secondary antibody incubation was carried out at room temperature for 1 h using horseradish-peroxidase-conjugated anti-rabbit IgG (Cell Signaling Technology, 7074). Washes were carried out with 0.1% Tween-20 Tris-buffered saline and blots were developed using enhanced chemiluminescence western blotting detection reagents (Cytiva, RPN2209).

### Protein expression and purification

For expressing ClC-3 in complex with T9A, the gene encoding human T9A (Q9P0T7) was synthesized subcloned into BacMam expression vector[50], in which a stop codon was introduced after the T9A sequence to ensure that T9A was not fused to a fluorophore with a linker containing a PreScission protease site. The N-terminal mCerulean-tagged ClC-3 construct described above was used for ClC-3/T9A complex expression. Equal amounts of the plasmid encoding N-terminal mCerulean-tagged ClC-3 and the plasmid encoding T9A without a fused fluorophore were mixed 1:3 (w/w) with PEI 25K (Polysciences) for 20 min and then used to transfect Expi293F cells (Gibco). Once the cell density reached $3 \times 10^6$ cells per ml, 1 mg of plasmid and 3 mg of PEI 25K were used to transfect a 1-L cell culture. For ClC-3 expression by itself, the plasmid encoding C-terminal mCerulean-tagged ClC-3 was used to transfect HEK293S GnTi$^-$ cells (American Type Culture Collection, CRL-3022) at a cell density of $2 \times 10^6$ cells per ml using the same protocol described above.

After 24-h incubation at 37 °C, valproic acid sodium salt (Sigma-Aldrich, P4543) was added to a final concentration of 2.2 mM and cells were allowed to grow at 37 °C for an additional 24 h before harvesting. Cell pellets were washed in PBS and flash-frozen in liquid nitrogen. Membrane proteins were solubilized in 2% LMNG (Anatrace, NG310), 0.2% cholesteryl hemisuccinate Tris salt (Anatrace, CH210), 20 mM HEPES pH 7.5 and 150 mM KCl supplemented with protease-inhibitor cocktail (1 mM PMSF, 2.5 mg ml$^{-1}$ aprotinin, 2.5 mg ml$^{-1}$ leupeptin and 1 mg ml$^{-1}$ pepstatin A) and DNase I. Solubilized proteins were separated by centrifugation at 135,557$g$ for 40 min at 4 °C, followed by binding to anti-GFP nanobody resin for 1.5 h at 4 °C, which was previously equilibrated with SEC buffer containing 0.02% glyco-diosgenin (Anatrace), 50 mM Tris-HCl (pH 8), 150 mM KCl and 2 mM DTT. Anti-GFP nanobody affinity chromatography was performed using 2–3 column volumes of washing with SEC buffer followed by overnight PreScission digestion and elution with the SEC buffer. The eluted protein sample was concentrated to a volume of 250 µl using Spin-X concentrators (100-kDa cutoff) (Corning, 431491), followed by centrifugation at 21,130$g$ for 15 min. Concentrated protein was further purified by SEC on Superose 6 Increase 10/300 GL (GE healthcare) in SEC buffer. Peak fractions were pooled and concentrated using Corning Spin-X concentrators (100-kDa cutoff) to the concentration specified below.

### EM sample preparation and data acquisition

For ClC-3 alone, 3 µl of 3 mg ml$^{-1}$ purified protein in the absence of any exogenous ligands was applied to glow-discharged Au 400-mesh Quantifoil R1.2/1.3 holey carbon grids (Quantifoil). The grids were plunged into liquid-nitrogen-cooled liquid ethane using an FEI Vitrobot Mark IV (FEI Thermo Fisher). The freezing process occurred at 4 °C under 100% humidity, with blotting times ranging from 2 to 3.5 s and a waiting time of 10 s. The grids were transferred to a 300-keV FEI Titan Krios G2 microscope equipped with a K3 summit direct electron detector (Gatan). Images were captured with SerialEM 4.2 (ref. 51) in

super-resolution mode at a magnification of ×29,000, corresponding to a pixel size of 0.413 Å. The dose rate during imaging was set at 15 e⁻ per pixel per s and the defocus range was from −0.7 to −2 μm. The image acquisition duration was 3 s, consisting of 0.05 subframes (total of 60 subframes), resulting in a cumulative dose of 66 e⁻ per Å².

For ClC-3 in complex with T9A, 3 μl of 5 mg ml⁻¹ purified protein in the absence of any exogenous ligands was applied to glow-discharged Au 400-mesh Quantifoil R1.2/1.3 holey carbon grids (Quantifoil) and then plunged into liquid-nitrogen-cooled liquid ethane with an FEI Vitrobot Mark IV (FEI Thermo Fisher). The sample was frozen at 4 °C with 100% humidity, using a blotting time of 2.5 s, blotting force of 2 and waiting time of 10 s. Grids were transferred to a 300-keV Titan Krios G4 microscope equipped with a Falcon4i direct detector and a SeletrisX energy filter (Thermo Fisher Scientific). The images were recorded with EPU 3.9.1 software at a pixel size of 0.725 Å. The dose rate was 11.51 e⁻ per pixel per s and the defocus range was −0.5 to −1.5 μm. Images were recorded for 2.93 s with a frame time of 0.003256 s (total 900 frames), corresponding to a total dose of 59.63 e⁻ per Å². The nominal magnification was ×165,000. The energy filter slit width was 10 eV.

### EM data processing

**ClC-3.** A total of 4,672 60-frame super-resolution videos (0.413 Å per pixel) of ClC-3 collected from one grid were gain-corrected, Fourier-cropped by two (0.826 Å per pixel) and aligned using whole-frame and local motion correction algorithms by cryoSPARC (version 3.2.0)[52]. Blob-based autopicking in cryoSPARC was implemented to select initial particles, resulting in stacks of 1,446,105 particles. False-positive selections and contaminants were removed by iterative rounds of heterogeneous classification using the initial three-dimensional (3D) reconstruction generated using the ab initio reconstruction in cryoSPARC, as well as several decoy classes generated from noise particles through ab initio reconstruction in cryoSPARC (version 4.2.1)[52]. After particle polishing in RELION (version 3.1.2)[53], a nonuniform refinement of the resulting particles yielded a reconstruction of 507,952 particles at 2.94 Å. Iterative rounds of heterogeneous classification followed by a $C_2$-symmetry nonuniform refinement in cryoSPARC (version 4.2.1) with local contrast transfer function (CTF) estimation and higher-order aberration correction resulted in a final reconstruction of 219,662 particles at 2.54 Å.

**ClC-3 in complex with T9A.** A total of 19,481 900-frame videos (0.725 Å per pixel) of ClC-3 in complex with T9A were collected from two grids using a Falcon4i direct detector with a SelectrisX energy filter at a slit width of 10 eV (Thermo Fisher Scientific). The videos were fractionated to 40 and aligned using whole-frame and local motion correction algorithms in cryoSPARC (version 4.2.1) using cryoSPARC Live[52]. Blob-based autopicking in cryoSPARC was implemented to select initial particles, resulting in stacks of 5,874,584 and 4,262,456 particles. A subset of particles with the best two-dimensional (2D) averages was selected after performing 2D classification in cryoSPARC. False-positive selections and contaminants of the subset particles were removed by iterative rounds of heterogeneous classification using the final human ClC-3 reconstruction, as well as several decoy classes generated from noise particles through ab initio reconstruction in cryoSPARC (version 4.2.1)[52]. After performing 2D classification of the resulting particle stacks, particles with the best 2D averages were then selected to train topaz picking in cryoSPARC (version 4.2.1)[54], resulting in stacks of 1,316,391 and 1,065,143 particles. False-positive selections and contaminants of the topaz-picked particles were excluded through iterative rounds of heterogeneous classification using the same inputs as above, resulting in stacks of 331,753 and 51,335 particles. False-positive selections and contaminants of the initial blob-based auto-picked particles were also removed by iterative rounds of heterogeneous classification using the same inputs, resulting in stacks of 408,736 and 342,430 particles. Duplicate particles from the blob-based auto-picked particles

and topaz-picked particles from the same micrograph dataset were removed. After performing reference-based motion correction of the duplicate-removed particles, the particles from the two datasets were combined. The combined particles were subjected to $C_2$-symmetry nonuniform refinement in cryoSPARC (version 4.2.1) with per-particle defocus and global CTF estimations, resulting in a reconstruction of 1,057,848 particles at 2.71 Å. The particle stack was then classified using 3D classification in cryoSPARC (version 4.2.1) with $C_1$ symmetry, using a focus mask that covers the density of T9A. A distinct class of ClC-3 dimer with no T9A density was identified. A $C_2$-symmetry nonuniform refinement of the class resulted in reconstruction of 148,161 particles at 2.90 Å. Particles of the remaining classes were further classified by iterative rounds of 3D classification with the T9A focus mask. Three classes with distinct T9A densities were identified. A $C_1$-symmetry nonuniform refinement of the first class, for which the CTD density of T9A was absent in both T9A structures and the LD density was absent in one of the two T9A structures, resulted in a reconstruction of 91,755 particles at 3.01 Å. A $C_2$-symmetry nonuniform refinement of the second class, for which LD and CTD densities of both T9A structures were present, resulted in a reconstruction of 94,011 particles at 2.86 Å. A $C_1$-symmetry nonuniform refinement of the third class, for which LD and CTD densities were present only in one of the two T9A structures, resulted in a reconstruction of 71,754 particles at 3.16 Å. Other classes consisted of mixed populations of poorly classified particles.

### Model building and coordinate refinement

**ClC-3.** The final reconstruction was subjected to density modification using the two unfiltered half-maps with a soft mask in PHENIX[55], yielding an improved density map at 2.45 Å. Human ClC-3 was manually built into the density-modified map in Coot (version 0.9.6)[56], followed by iterative rounds of real-space refinement in PHENIX (version 1.21.1-5286)[57] and manual rebuilding in Coot (version 0.9.6)[56].

**ClC-3 in complex with T9A.** The final reconstructions were subjected to density modification using the two unfiltered half-maps with a soft mask in PHENIX[55]. The structure of human ClC-3 alone was docked into the maps and manually rebuilt in Coot (version 0.9.6) to fit the density[56]. Human T9A was then manually built into the density-modified map. The models were refined by iterative rounds of real-space refinement in PHENIX (version 1.21.1-5286)[57] and manual rebuilding in Coot (version 0.9.6)[56].

### Molecular biology

Expression plasmids encoding human ClC-3a or ClC-5 complementary DNA (cDNA), both with Venus fused to the N terminus, and mouse *Tmem9* cDNA were used for the vacuolization assay[29]. ClC-3 or Tmem9 point mutants were generated with the Agilent QuikChange kit (Agilent, 200523). Oligonucleotides used in the site-directed mutagenesis were designed using Agilent's QuickChange primer design tool (https://www.agilent.com/store/primerDesignProgram.jsp). Oligonucleotides used for mutagenesis are listed in Supplementary Table 1. All clones were verified by sequencing the entire open reading frame.

### CLC vacuolization assay

HeLa WT cells (Leibniz-Institut DSMZ) or Hela *CLCN7*-knockout (KO) cells[58] were cultured on glass coverslips in complete medium (DMEM + 10% FBS + 1% penicillin–streptomycin) at 37 °C and 5% CO₂. Cells were transfected using JetPRIME (Polyplus) as the transfection reagent with plasmids expressing Venus–ClC-3a WT or Venus–ClC-3a mutants including the *td* mutant in which the 'proton glutamate' is substituted by alanine (corresponding to E281A in the short ClC-3a isoform and E339A in the long isoform used for structural analysis) or Venus–ClC-5 WT together with WT or mutant T9A, under the control of the cytomegolvirus promoter. In some experiments, T9A was substituted by CD4 as a control.

Then, 48 h after transfection, cells growing on coverslips were processed for immunocytochemistry. In experiments investigating the effect of PtdIns(3,5)P$_2$, this was preceded by a 3–4-h incubation with PIKfyve inhibitors (either 100 nM apilimod (Sigma-Aldrich, SML2974) or 1 μM YM-201636 (Merck, 371942-69-7) (both dissolved in 0.005% DMSO) in complete medium at 37 °C with 5% CO$_2$). In these experiments, 0.005% DMSO in complete medium served as a vehicle control. Cells were then washed twice in cold PBS and fixed with chilled methanol for 10–15 min. Cells were extensively washed with PBS, blocked and permeabilized with 0.1% saponin in 3% sterile-filtered goat serum (Pan-Biotech, P30-1002) and 2% BSA at room temperature for 1 h. Cells were then incubated overnight at 4 °C with primary antibodies (chicken anti-GFP antibody (1:500; Aves Lab, GFP-1020), guinea pig anti-T9A (T9AC2) (1:1,000; Pineda Antibody Service[29]) and mouse anti-Lamp-2 (H4B4) (1:500; Abcam, Ab25631) in 3% BSA and 0.05% saponin. Next, cells were extensively washed in 0.05% saponin–PBS, incubated with secondary antibodies coupled to different fluorophores (goat anti-chicken Alexa 488, A11039, 1:1,000; goat anti-guinea pig Alexa 555, A21435, 1:1,000; goat anti-mouse Alexa 633, A21052, 1:1,000; all from Invitrogen) and 1 μg ml$^{-1}$ DAPI at room temperature for 1 h, then washed and mounted on slides using Fluoromount-G (SouthernBiotech) and allowed to dry. Images were acquired using an LSM880 Zeiss confocal microscope using ×40 water-immersion or ×63 oil-immersion (numerical aperture 1.4) lenses.

Each condition (mutants or inhibitor treatments) was examined in a minimum of three biological replicates. To ensure unbiased sampling, 8–16 images were randomly acquired using a ×40 water-immersion objective with a Zeiss LSM880 confocal microscope for each individual experiment. To restrict the analysis to cells coexpressing both proteins, cells that displayed above-background fluorescence in both the Venus and the T9 channel (about 20–25% of total cells) were manually selected for further analysis. Automated vesicle size determination with available ImageJ plugins was not possible because the software could not distinguish individual vesicles when occurring in clusters, which are frequently observed with ClC-3-enlarged vesicles. Therefore, the cross-sectional areas of individual ClC-3-containing vacuoles were calculated with ImageJ (National Institutes of Health (IH)) from oval or elliptical regions of interest that were manually drawn on the perimeter (indicated by Venus–ClC-3 expression) of individual vesicles.

Enlarged vacuoles were defined by setting a minimum size threshold (cross-sectional area > 1.0 μm$^2$). On the basis of the number and size of vacuoles present in each cell, we qualitatively classified cells into three distinct categories: (1) cells containing small vesicles or punctae, with area < 1.0 μm$^2$, observed in Venus–ClC-3 coexpressed with WT T9A; (2) cells containing five or more vacuoles with cross-sectional areas between 1.0 and 1.5 μm$^2$; and (3) cells containing five or more vacuoles with cross-sectional areas greater than 1.5 μm$^2$. The percentage of cells falling into the different vacuole size categories was calculated and plotted as a bar plot. This analysis showed that enlarged vacuoles were only observed when either ClC-3 or T9A carried mutations or PIKfyve was inhibited and revealed differences between the degree of disinhibition in a semiquantitative manner. For experiments with PIKfyve inhibitors, only cells containing ≥5 vacuoles with a cross-sectional area > 1.5 μm$^2$ were considered, as the effect appeared to be all or none.

## Statistics and reproducibility

All FSEC experiments were independently performed three times, with the results being similar. All vacuolization experiments were independently performed at least two times, with the results being similar. No statistical methods were used to predetermine sample size. All vacuolization experiments were performed at least two times, with the results being similar. The images presented in Fig. 4c,d are representative of three independent experiments (biological replicates) with each replicate investigated in two coverslips. The images presented in Fig. 5c,d,g are representative of three independent experiments (biological replicates). The images presented in Fig. 5e,h are representative of five independent experiments (biological replicates). The image presented in Fig. 5f is representative of six independent experiments (biological replicates). The images presented in Extended Data Fig. 8a–e are representative of the images collected across two independent experiments (biological replicates). To restrict the vacuolization analysis to cells coexpressing both proteins, cells that displayed above-background fluorescence in both the Venus and the T9 channel (about 20–25% of total cells) were manually selected for further analysis. The vacuolization experiments were not randomized and the Investigators were not blinded to allocation during experiments and outcome assessment. Cryo-EM images were collected from 1–2 grids per sample condition. The number of collected images for each condition is indicated in Extended Data Figs. 2 and 3. Particle images were excluded using 2D and 3D classification approaches.

## Figures

Figures were prepared with PyMol 2.5.3. (www.pymol.org), ChimeraX (version 1.5)[59], GraphPad Prism 9 (www.graphpad.com), Clustal Omega (version 1.2.3)[60] and MOLE[61].

## Reporting summary

Further information on research design is available in the Nature Portfolio Reporting Summary linked to this article.

## Data Availability

Cryo-EM maps were deposited to the EM Data Bank under accession codes EMD-47070 (ClC-3), EMD-47066 (ClC-3/noT9A), EMD-47067 (ClC-3/T9A, T9A protomers A and B: complete), EMD-47068 (ClC-3/T9A, T9A protomer A: no CD, T9A protomer B: no LD, no CD) and EMD-47069 (ClC-3/T9A, T9A protomer A: complete, T9A protomer B: no LD, no CD). Atomic coordinates were deposited to the PDB under accession codes 9DO0 (ClC-3), 9DNW (ClC-3/noT9A), 9DNX (ClC-3/T9A, T9A protomers A and B: complete), 9DNY (ClC-3/T9A, T9A protomer A: no CD, T9A protomer B: no LD, no CD) and 9DNZ (ClC-3/T9A, T9A protomer A: complete, T9A protomer B: no LD, no CD). The atomic coordinates of previously published structures of bovine ClC-K (PDB 5TQQ), human ClC-2 (PDB 8TA4), human ClC-6 (PDB 8JPJ) and human ClC-7/OSTM1 complex (PDB 7JM7) were used in this study. All other data supporting the findings of this study are available from the corresponding authors on reasonable request. Source data are provided with this paper.

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

## Acknowledgements

We thank M. J. de la Cruz for help with data acquisition, the Memorial Sloan Kettering Cancer Center HPC group for assistance with data processing and the members of the laboratories for comments on the paper. This study was supported in the R.K.H. lab in part by the National Institute of General Medical Sciences (R01-GM141553) and NIH National Cancer Institute Cancer Center Support Grant (P30-CA008748). This study was supported in the T.J.J. lab in part by the European Research Council Advanced Grant 740537 (VolSignal) and the Deutsche Forschungsgemeinschaft (DFG) JE164/12-2 and under Germany's Excellence Strategy (EXC-2049, Project ID 390688087, Neurocure). This study was supported in the B.F. lab in part by grants of the DFG (FA 332/15-1, 16-1 and 21-1 and the CRC/TRRs 1453 and 152). M.S. is supported by the Walter Benjamin Program of the DFG.

## Author contributions

Conceptualization, M.S., Y.S., T.J.J. and R.K.H. Methodology, M.S., Y.S., R.P.-C., V.V., S.K., U.S., B.F., T.J.J. and R.K.H. Formal analysis, M.S., Y.S., R.P.-C., V.V., S.K., U.S., B.F., T.J.J. and R.K.H. Investigation, M.S., Y.S., R.P.-C., V.V., S.K., U.S., B.F., T.J.J. and R.K.H. Writing—original draft, M.S., Y.S. and R.K.H. Funding acquisition, M.S., B.F., T.J.J. and R.K.H.

## Funding

V. (FMP).

## Competing interests

R.H. is a consultant for F. Hoffmann-La Roche. The other authors declare no competing interests.

## Additional information

**Extended data** is available for this paper at https://doi.org/10.1038/s41594-025-01617-2.

**Correspondence and requests for materials** should be addressed to Thomas J. Jentsch or Richard K. Hite.

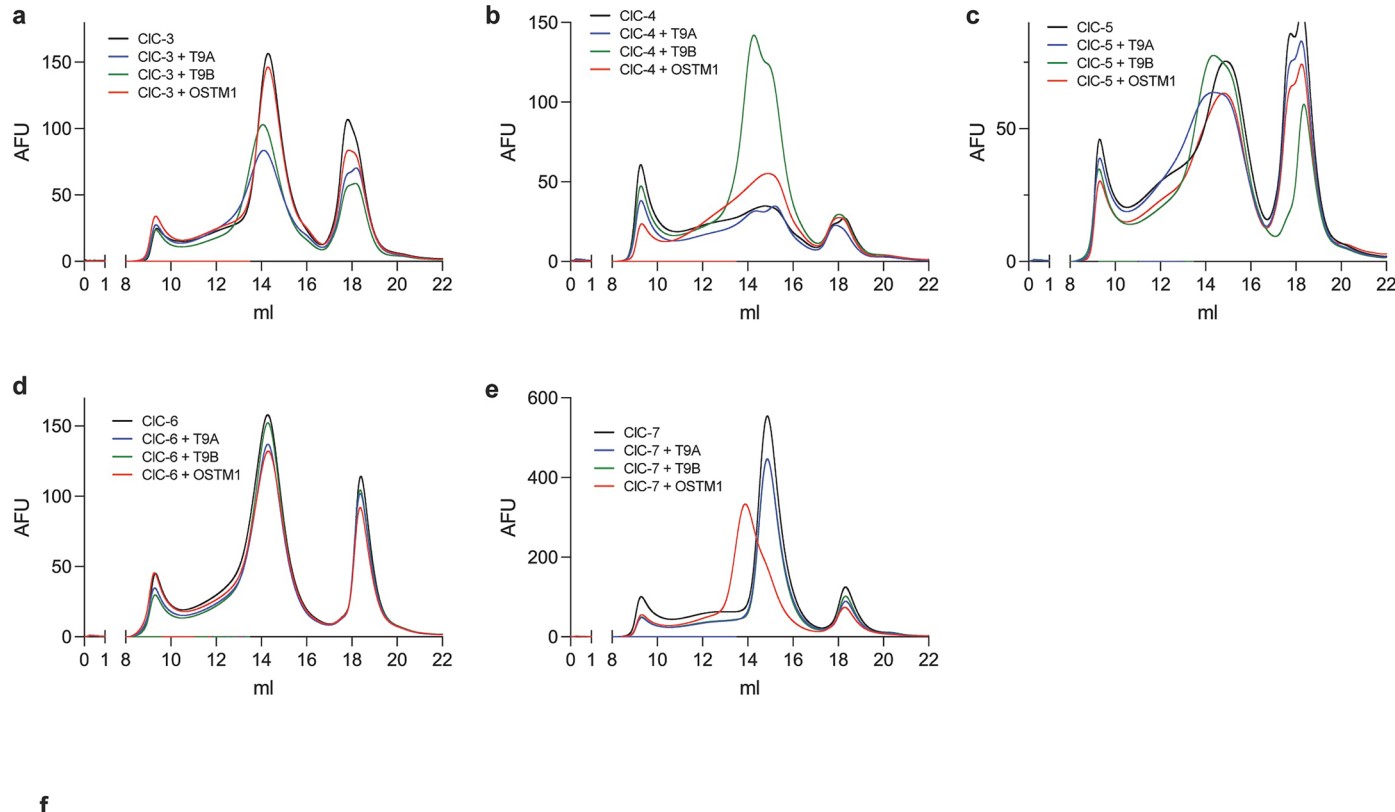

**Extended Data Fig. 1 | T9A and T9B form complexes with ClC-3, ClC-4 and ClC-5.**
(**a**) FSEC profiles of mCerulean-ClC-3 (black), mCerulean-ClC-3 co-expressed with mVenus-T9A (blue), mCerulean-ClC-3 co-expressed with mVenus-T9B (green), and mCerulean-ClC-3 co-expressed with mVenus-OSTM1 (red).
(**b**) FSEC profiles of mCerulean-ClC-4 (black), mCerulean-ClC-4 co-expressed with mVenus-T9A (blue), mCerulean-ClC-4 co-expressed with mVenus-T9B (green), and mCerulean-ClC-4 co-expressed with mVenus-OSTM1 (red).
(**c**) FSEC profiles of mCerulean-ClC-5 (black), mCerulean-ClC-5 co-expressed with mVenus-T9A (blue), mCerulean-ClC-5 co-expressed with mVenus-T9B (green), and mCerulean-ClC-5 co-expressed with mVenus-OSTM1 (red).

(**d**) FSEC profiles of mCerulean-ClC-6 (black), mCerulean-ClC-6 co-expressed with mVenus-T9A (blue), mCerulean-ClC-6 co-expressed with mVenus-T9B (green), and mCerulean-ClC-6 co-expressed with mVenus-OSTM1 (red).
(**e**) FSEC profiles of mCerulean-ClC-7 (black), mCerulean-ClC-7 co-expressed with mVenus-T9A (blue), mCerulean-ClC-3 co-expressed with mVenus-T9B (green), and mCerulean-ClC-7 co-expressed with mVenus-OSTM1 (red). Expi293F cells were transfected with different DNA constructs. After protein purification, the fluorescence of mCerulean-tagged protein was monitored at an excitation/emission wavelength of 433/475 nm, respectively. (**f**) Twin-strep tagged T9A co-immunoprecipitated with ClC-3,4, and -5, but not ClC-6.

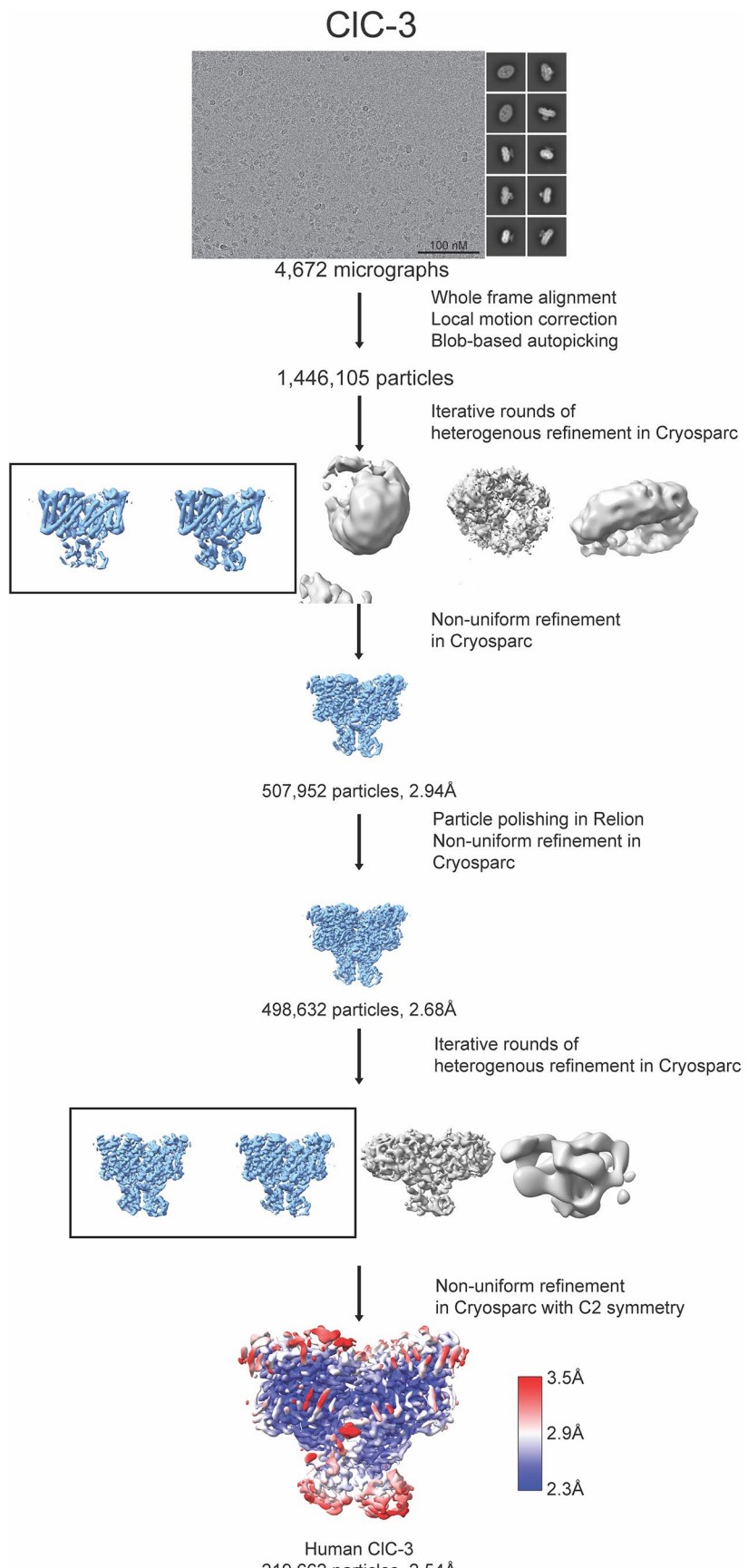

**Extended Data Fig. 2 | Cryo-EM analysis of ClC-3.** Workflow of cryo-EM image analysis for ClC-3 with representative micrograph, 2D averages, 3D averages and final reconstruction colored by local resolution.

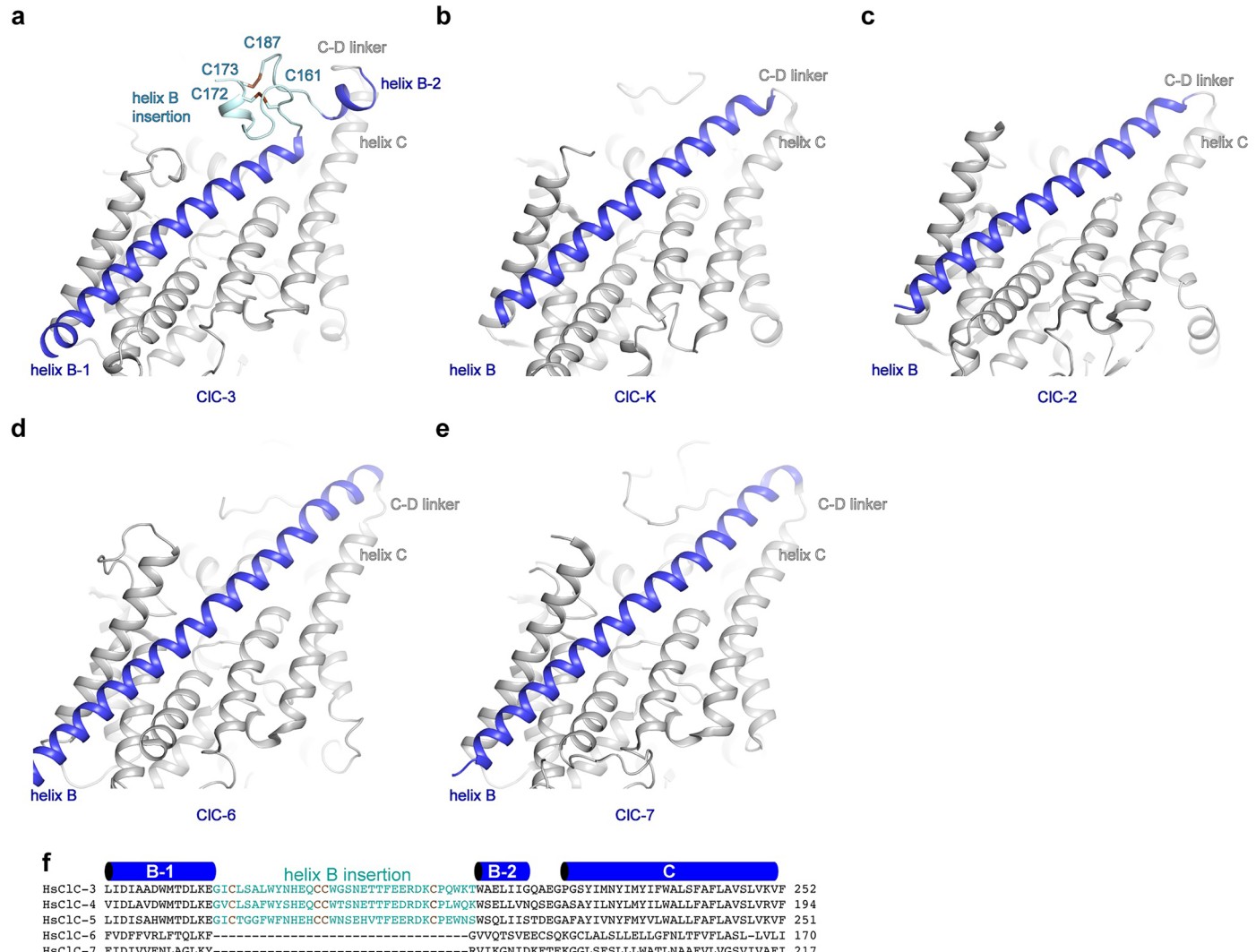

**Extended Data Fig. 3 | Conformation of helix B in mammalian CLC structures.**
(**a-e**) Cartoon representation of TMD of human ClC-3 (**a**), bovine ClC-K (**b**, PDB 5TQQ), human ClC-2 (**c**, PDB 8TA4), human ClC-6 (**d**, PDB 8JPJ) and human ClC-7 (**e**, PDB 7JM7). Helix B is colored in blue, the helix B insertion in ClC-3 is colored in light blue and all other residues are colored in grey. Residues in ClC-3 that form disulfide bonds are shown as sticks. (**f**) Sequence alignment of helix B of human ClC-3, ClC-4, ClC-5, ClC-6, and ClC-7. Secondary structural elements and residues that form disulfide bonds in ClC-3 are highlighted.

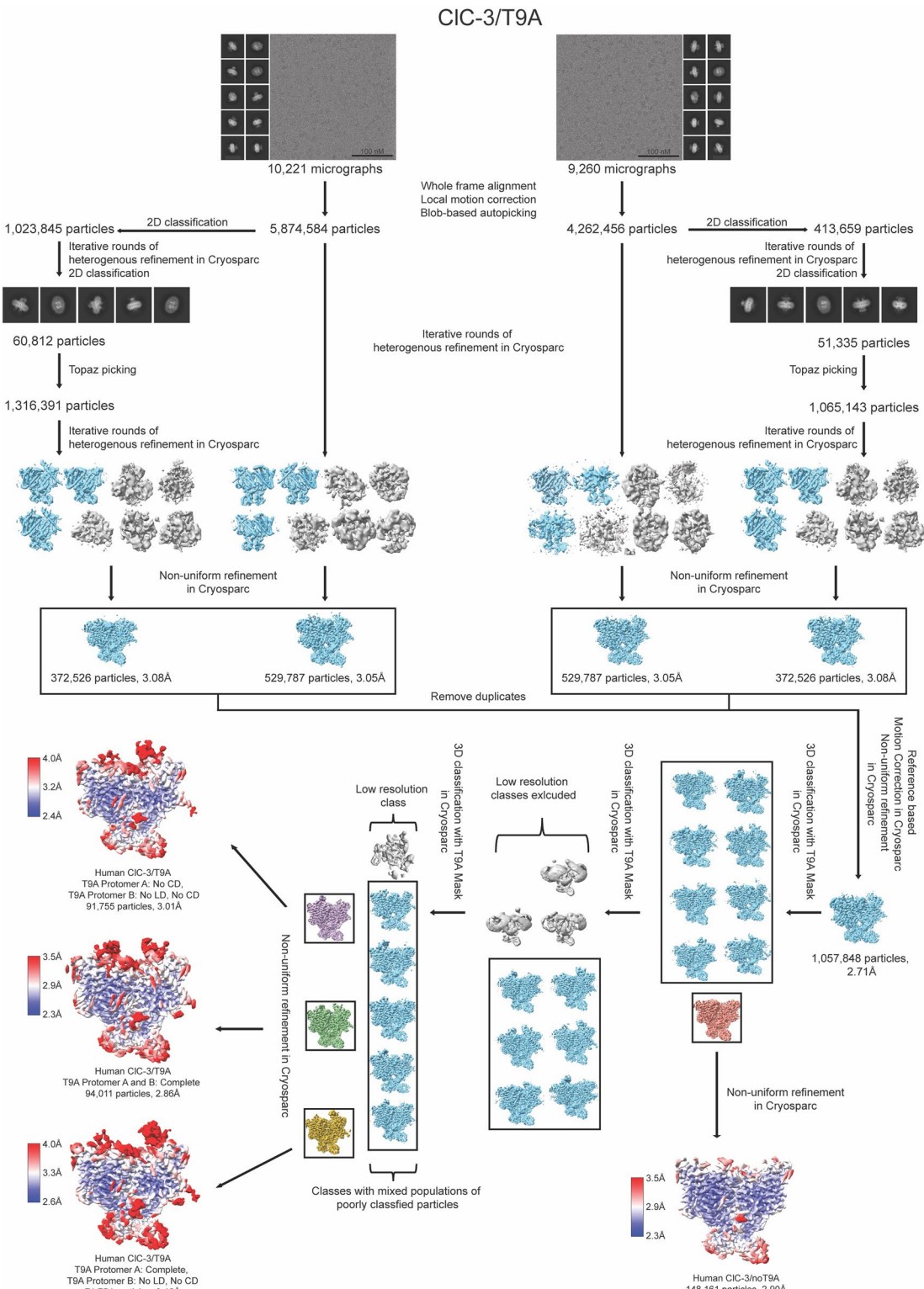

**Extended Data Fig. 4 | Cryo-EM analysis of ClC-3 in complex with T9A.** Workflow of cryo-EM image analysis for ClC-3 in the presence of T9A with representative micrograph, 2D averages, 3D averages and final reconstruction colored by local resolution.

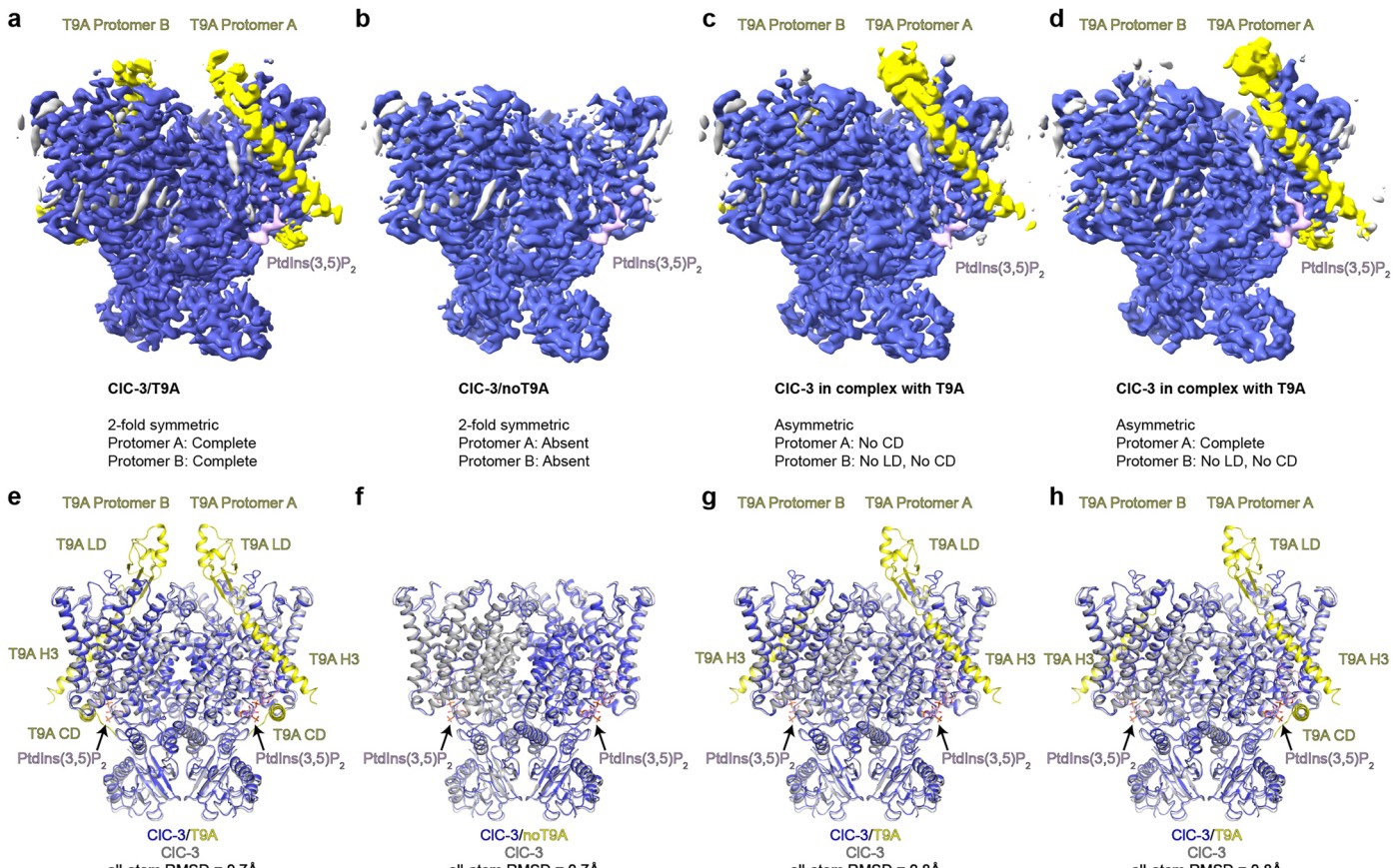

**Extended Data Fig. 5 | Structures of ClC-3 in the presence of T9A. (a-d)** Cryo-EM maps of ClC-3/T9A (**a**), ClC-3/noT9A (**b**) and two classes of ClC-3 in complex with T9A in which the T9A protomers are only partially ordered (**c,d**), colored by subunit. Densities corresponding to ClC-3 and PI(3,5)P₂ are observed in all classes and are colored blue and pink, respectively. Densities corresponding to T9A are observed in all classes except for ClC-3/noT9A and are colored yellow. (**e-h**) Superposition of the atomic models of ClC-3 with ClC-3/T9A (**e**), ClC-3

with ClC-3/noT9A (**f**) ClC-3 with the two classes of ClC-3 in complex with T9A in which the T9A protomers are only partially ordered (**g,h**). The ClC-3 only model is colored grey. ClC-3 from the ClC-3/T9A or ClC-3/T9A models is colored blue, T9A is colored yellow and PI(3,5)P₂ is colored pink. Root mean squared deviation (RMSD) calculations were performed using all overlapping ClC-3 atoms in the ClC-3 only structures and the ClC-3/T9A or ClC-3/noT9A structures.

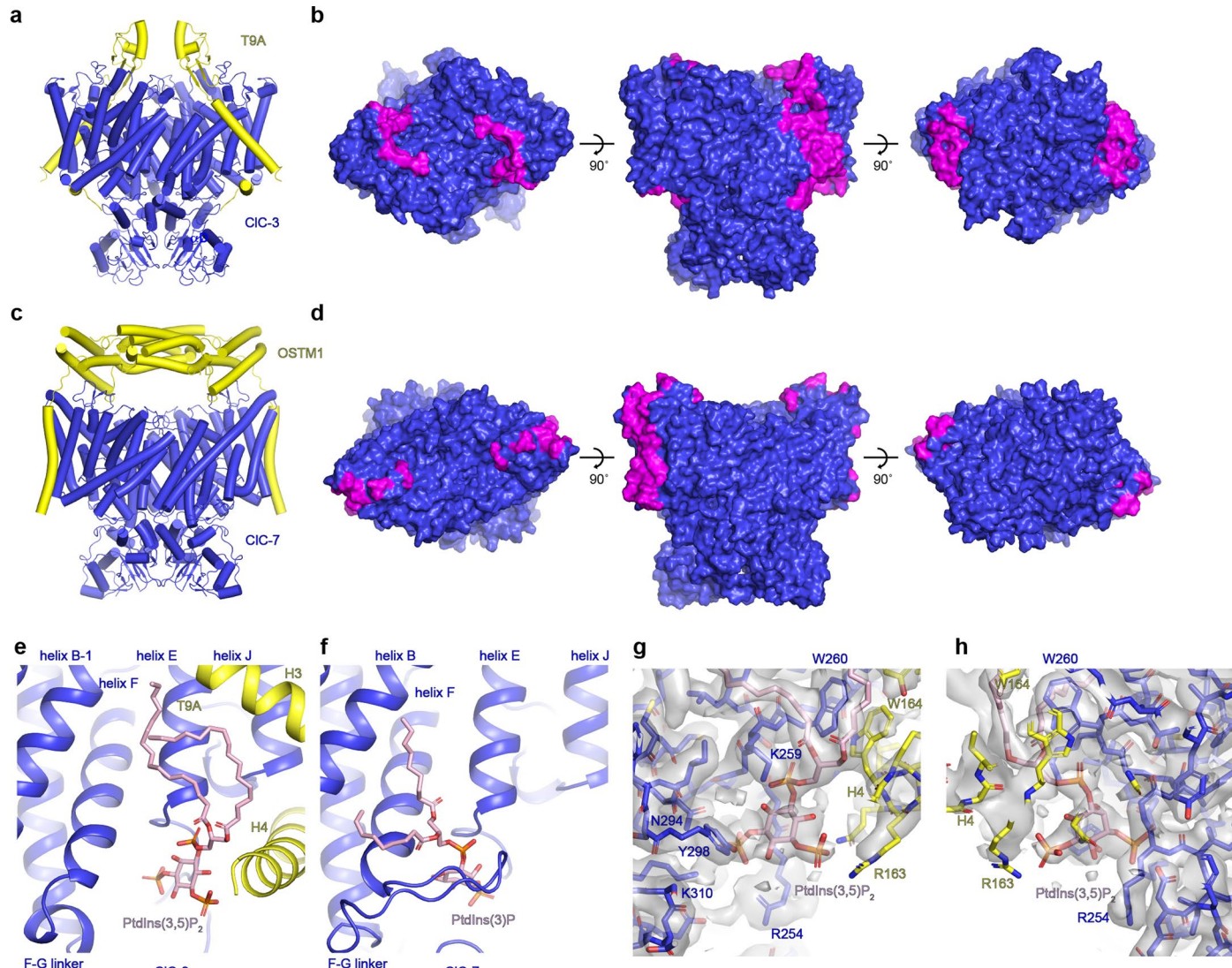

**Extended Data Fig. 6 | Comparison of ClC-3/T9A and ClC-7/OSTM1 complexes.** (**a,c**) Structures of ClC-3/T9A (**a**) and ClC-7/OSTM1 (**c**, PDB: 7JM7), colored by subunit. (**b,d**) Surface representation of ClC-3 (**b**) and ClC-7 (**d**), viewed from the lumen (left), from within the plane of the membrane (middle) and cytosol (right).

Residues that interact with T9A or OSTM1 are colored in magenta. (**e**) PI(3,5)P$_2$ binding site in ClC-3/T9A. (**f**) PI(3)P binding site in ClC-7/OSTM1 (PDB 7JM7). (**g-h**) Two views of the cryo-EM density map in the vicinity of the PI(3,5)P$_2$ binding site in ClC-3/T9A.

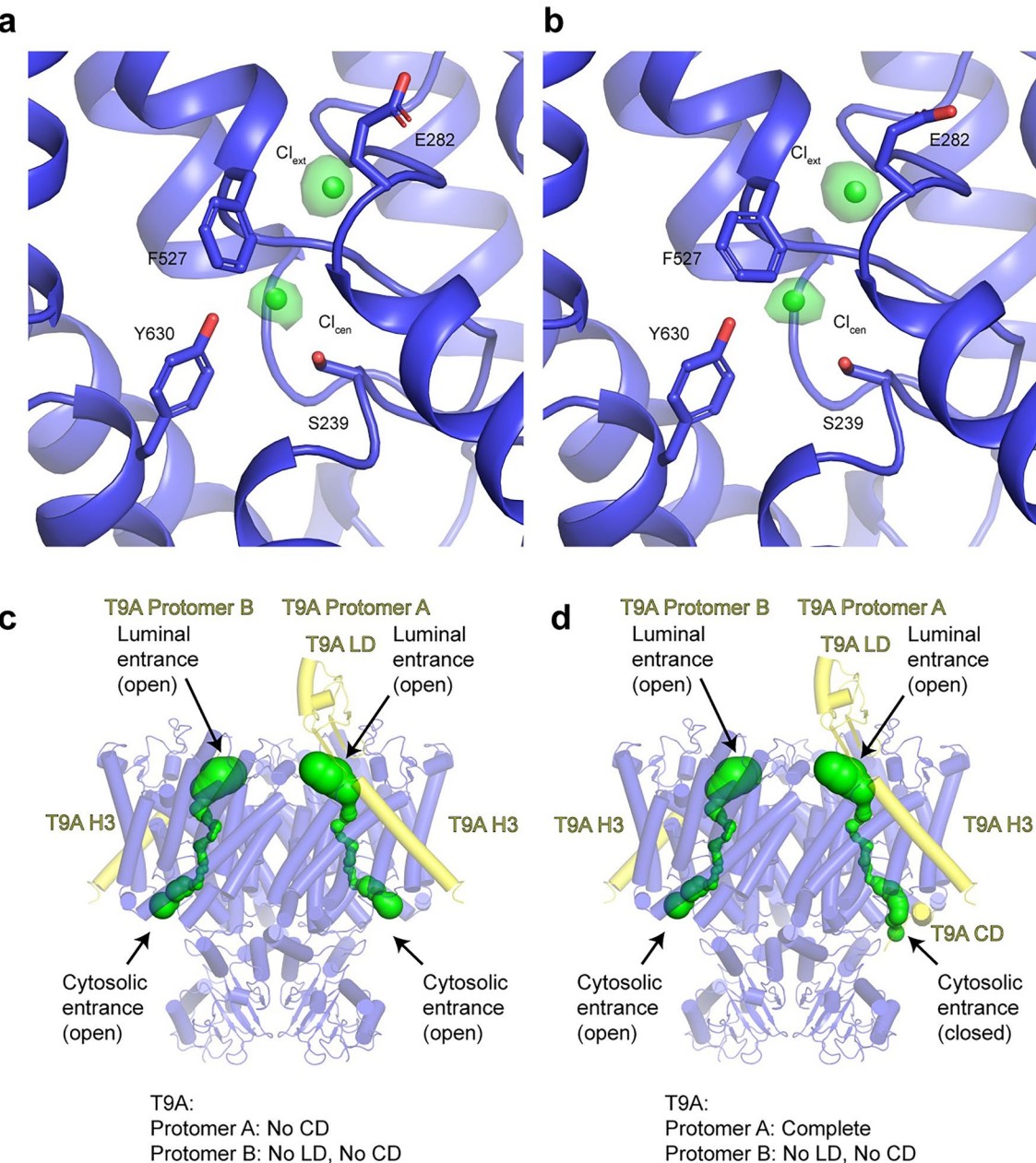

**Extended Data Fig. 7 | Cl⁻ ion pathways of ClC-3 in the presence of T9A.** (**a-b**) Cl⁻-binding sites in the Cl⁻ ion pathway of one protomer of ClC-3/T9A (**a**) or ClC-3/noT9A (**b**). Densities are shown as green isosurfaces and contoured at 4.0 σ. (**c-d**) Cl⁻ ion pathways of two classes of ClC-3 in complex with T9A in which the T9A protomers are only partially ordered, depicted as green surfaces.

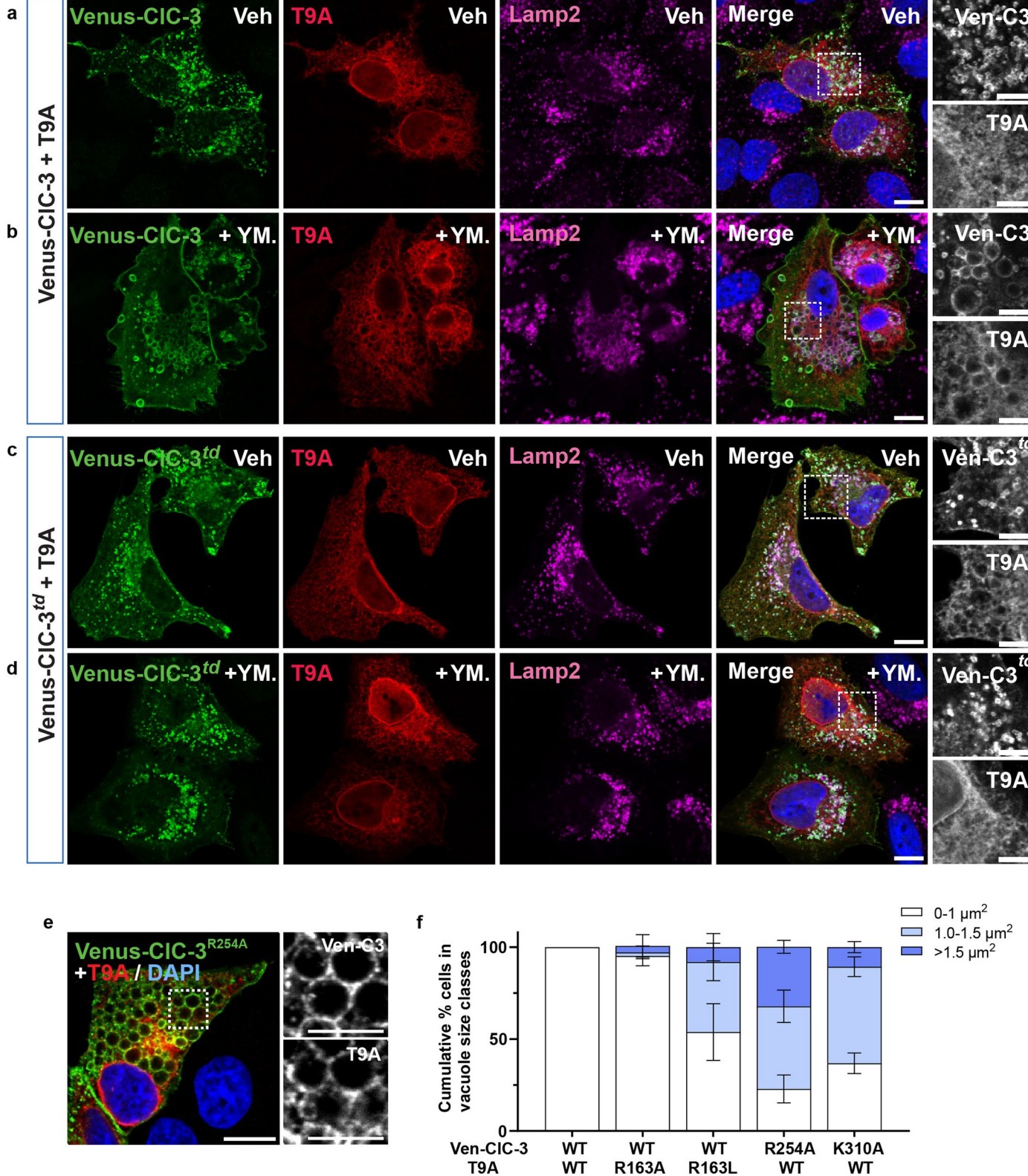

**Extended Data Fig. 8 | See next page for caption.**

**Extended Data Fig. 8 | PI(3,5)P$_2$ and T9A are required to fully inhibit ClC-3.**
(**a-d**) T9A-mediated inhibition of ClC-3a-dependent vacuolization is relieved by YM201636. (**c-d**) Co-expression of T9A with transport-deficient ClC-3a$^{td}$ mutant does not support YM201636-induced vacuolization. Representative confocal images of ClC-7 deficient HeLa cells co-expressing Venus-ClC-3a (**a-b**) or Venus-ClC-3a$^{td}$ (**c-d**) (in green) with T9A (red), untreated (**a,c**) or treated with 1 μM YM201636 for 4 hours (**b,d**). Merged images at right. Zoomed-in view of boxed region is shown at top right for Venus-ClC-3a WT (Ven-C3) or td mutant (Ven-C3$^{td}$) and bottom right for T9A. Scale bars: 10 μm, and 5 μm for enlarged areas. (**e**) Representative confocal images of ClC-3a-dependent vacuolization in cells co-expressing Venus-ClC-3a$^{R254A}$ and T9A. Venus-ClC-3a$^{R254A}$ is shown in green, T9A is shown in red and DAPI is shown in blue. Zoomed in view of boxed region is shown at top right for Venus-ClC-3a$^{R254A}$ (Ven-C3) and bottom right for T9A. Scale bar: 10 μm, 5 μm for enlarged areas. (**f**) Cumulative percentage of cells displaying *punctae* (<1 μm$^2$), ≥5 moderate size vacuoles (1–1.5 μm$^2$) per cell, or ≥5 large vacuoles (>1.5 μm$^2$) per cell. Vacuolization was investigated in fixed immunostained cells co-expressing Venus-ClC-3a with T9A, Venus-ClC-3a with T9A$^{R163A}$, Venus-ClC-3a with T9A$^{R163L}$, Venus-ClC-3a$^{R254A}$ with T9A, and Venus-ClC-3a$^{K310A}$ with T9A, as a semiquantitative measure of ClC-3 transport inhibition by T9A or its mutants. Data are shown as mean ± SD of 3 biological replicates. Numerical data are given in Source Data provided with this paper.

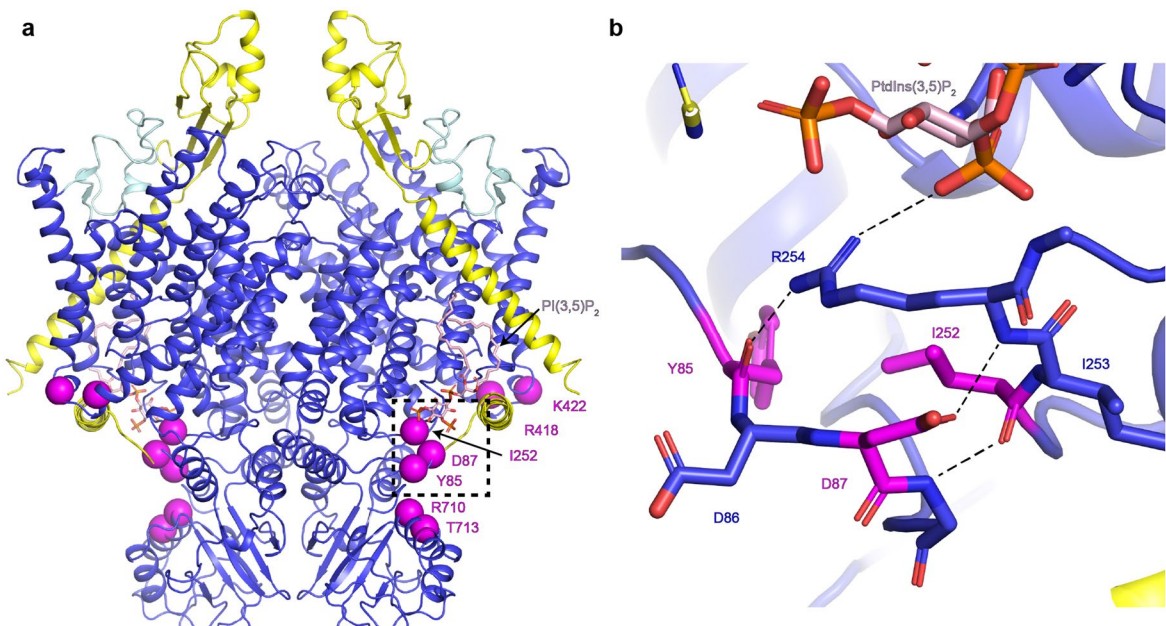

**Extended Data Fig. 9 | Disease-associated mutations in ClC-3/T9A. (a)** Residues whose mutation are associated with disease and weaken transport inhibition by T9A[29] are depicted as magenta spheres. Region inside dashed box is highlighted in **b**. **(b)** Network of interactions that stabilize the interface between ClC-3 and PI(3,5)P$_2$. Residues whose mutation are associated with human diseases are highlighted in magenta.

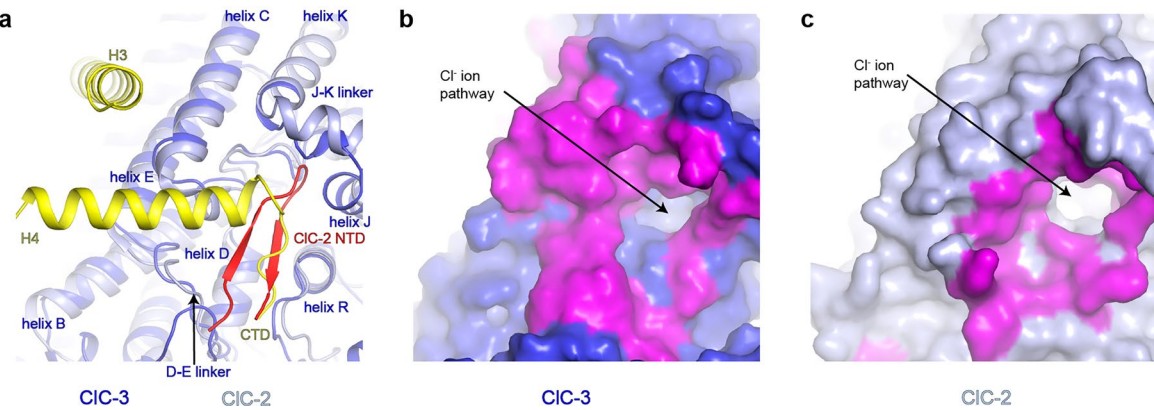

**Extended Data Fig. 10 | Comparison of the cytoplasmic entrances to the Cl⁻ ion pathways of ClC-3/T9A and ClC-2.** (**a**) Superposition of the region surrounding the cytosolic entrances to the Cl⁻ ion pathways of ClC-3/T9A and ClC-2 (PDB 8TA4). ClC-3 is colored blue, T9A is colored yellow, ClC-2 is colored light blue, and the ClC-2 NTD is colored red. (**b-c**) Surface representation of the region surrounding the cytosolic entrances to the Cl⁻ ion pathways of ClC-3 (**b**) and ClC-2 (**c**), with residues that interact with T9A or the N-terminus of ClC-2 colored in magenta.

Jentsch, Thomas J

# Reporting Summary

## Statistics

For all statistical analyses, confirm that the following items are present in the figure legend, table legend, main text, or Methods section.

| n/a | Confirmed | |
|---|---|---|
| ☐ | ☒ | The exact sample size (*n*) for each experimental group/condition, given as a discrete number and unit of measurement |
| ☐ | ☒ | A statement on whether measurements were taken from distinct samples or whether the same sample was measured repeatedly |
| ☐ | ☒ | The statistical test(s) used AND whether they are one- or two-sided<br>*Only common tests should be described solely by name; describe more complex techniques in the Methods section.* |
| ☒ | ☐ | A description of all covariates tested |
| ☒ | ☐ | A description of any assumptions or corrections, such as tests of normality and adjustment for multiple comparisons |
| ☐ | ☒ | A full description of the statistical parameters including central tendency (e.g. means) or other basic estimates (e.g. regression coefficient) AND variation (e.g. standard deviation) or associated estimates of uncertainty (e.g. confidence intervals) |
| ☐ | ☒ | For null hypothesis testing, the test statistic (e.g. *F*, *t*, *r*) with confidence intervals, effect sizes, degrees of freedom and *P* value noted<br>*Give P values as exact values whenever suitable.* |
| ☒ | ☐ | For Bayesian analysis, information on the choice of priors and Markov chain Monte Carlo settings |
| ☒ | ☐ | For hierarchical and complex designs, identification of the appropriate level for tests and full reporting of outcomes |
| ☒ | ☐ | Estimates of effect sizes (e.g. Cohen's *d*, Pearson's *r*), indicating how they were calculated |

*Our web collection on statistics for biologists contains articles on many of the points above.*

## Software and code

Policy information about availability of computer code

| Data collection | SerialEM 4.2, EPU 3.9.1, CryoSparc Live v4.2.1 |
|---|---|
| Data analysis | CryoSparc v3.2.0, CryoSparc v4.2.1, PyEM (https://doi.org/10.5281/zenodo.3576630), Clustal Omega (1.2.3) Relion v3.1.3, Phenix v1.21.1-5286, Coot 0.9.6, PyMol (Schrodinger, LLC. 2010. The PyMOL Molecular Graphics System, Version 2.5.3), ChimeraX 1.5, GraphPad Prism 9, MOLE, ImageJ |

For manuscripts utilizing custom algorithms or software that are central to the research but not yet described in published literature, software must be made available to editors and reviewers. We strongly encourage code deposition in a community repository (e.g. GitHub). See the Nature Portfolio guidelines for submitting code & software for further information.

## Data

Policy information about availability of data

All manuscripts must include a data availability statement. This statement should provide the following information, where applicable:
- Accession codes, unique identifiers, or web links for publicly available datasets
- A description of any restrictions on data availability
- For clinical datasets or third party data, please ensure that the statement adheres to our policy

Cryo-EM maps have been deposited in the EMDB under accession codes EMD-47070 [https://www.ebi.ac.uk/emdb/EMD-47070] (ClC-3), EMD-47066 [https://www.ebi.ac.uk/emdb/EMD-47066] (ClC-3/noT9A), EMD-47067 [https://www.ebi.ac.uk/emdb/EMD-46067] (ClC-3/T9A, T9A Protomer A and B: Complete),

## Human research participants

| | |
|---|---|
| Reporting on sex and gender | N/A |
| Population characteristics | N/A |
| Recruitment | N/A |
| Ethics oversight | N/A |

Note that full information on the approval of the study protocol must also be provided in the manuscript.

# Field-specific reporting

Please select the one below that is the best fit for your research. If you are not sure, read the appropriate sections before making your selection.

☒ Life sciences ☐ Behavioural & social sciences ☐ Ecological, evolutionary & environmental sciences

For a reference copy of the document with all sections, see nature.com/documents/nr-reporting-summary-flat.pdf

# Life sciences study design

All studies must disclose on these points even when the disclosure is negative.

| | |
|---|---|
| Sample size | Sample sizes were not predetermined with statistical means but based on standard numbers in the field. Cryo-EM sample size was determined by the availible microscope time. The number of images collected is indicated in Extended Data Figure 2. |
| Data exclusions | Cryo-EM images were excluded from the data set when they showed evidence of high drift or poor CTF fits. Individual particles were excluded by 2D and 3D classification as is the standard in the field of single-particle cryo-EM analysis. The selection of particles is shown in Extended Data Figure 2.<br>For the CLC vacuolization assay, only cells which displayed above-background fluorescence in both the Venus- and the T9 channel (about 20-25% of total cells) were manually selected for further analysis. |
| Replication | Cryo-EM: Each condition was imaged from one or two grids, with the results being similar.<br>All CLC vacuolization experiments were independently performed two times, with the results being similar. |
| Randomization | Cryo-EM particles were randomized during image processing to calculate FSC curves.<br>No randomization was done for vacuolization assays. |
| Blinding | Blinding was not performed as cryo-EM image analysis requires careful evaluation at step of the image processing workflow to ensure that high-quality reconstructions are obtained.<br>No blinding was performed for vacuolization assays, but fields randomly chosen. Several fields were imaged per preparation and by different researchers to avoid acquisition bias. |

# Reporting for specific materials, systems and methods

We require information from authors about some types of materials, experimental systems and methods used in many studies. Here, indicate whether each material, system or method listed is relevant to your study. If you are not sure if a list item applies to your research, read the appropriate section before selecting a response.

## Materials & experimental systems

| n/a | Involved in the study |
|---|---|
| ☐ | ☒ Antibodies |
| ☐ | ☒ Eukaryotic cell lines |
| ☒ | ☐ Palaeontology and archaeology |
| ☒ | ☐ Animals and other organisms |
| ☒ | ☐ Clinical data |
| ☒ | ☐ Dual use research of concern |

## Methods

| n/a | Involved in the study |
|---|---|
| ☒ | ☐ ChIP-seq |
| ☒ | ☐ Flow cytometry |
| ☒ | ☐ MRI-based neuroimaging |

## Antibodies

| | |
|---|---|
| Antibodies used | chicken anti-GFP antibody (1: 500; Cat# GFP-1020, Aves Lab)<br>guinea pig anti-T9A (T9AC2) (1:1000; Pineda Antibody Service, Berlin)<br>mouse anti-Lamp-2 (H4B4) (1:500; Cat. Ab25631, Abcam)<br>Goat anti-chicken Alexa 488 Cat. A11039<br>Goat anti-guinea pig Alexa 555 Cat. A21435<br>goat anti-mouse Alexa 633 Cat. A21052, Invitrogen |
| Validation | Validation for anti-T9A antibody is presented in reference 29. |

## Eukaryotic cell lines

Policy information about cell lines and Sex and Gender in Research

| | |
|---|---|
| Cell line source(s) | HEK293S GnTl- (ATCC CRL-3022)<br>Expi293F (Gibco)<br>HeLa  (Leibniz-Institut DSMZ- Deutsche Sammlung von Mikroorganismen und Zellkulturen GmbH, Germany)<br>HeLa ClC-7 KO (reference 58) |
| Authentication | Cells were authenticated by the Deutsche Sammlung von Mikroorganismen und Zellkulturen, Germany |
| Mycoplasma contamination | Cells were regularly tested for contamination by PCR |
| Commonly misidentified lines<br>(See ICLAC register) | No commonly misidentified cell lines were used in this study. |

