## [Peer Review File · Nature Structural & Molecular Biology]

Structural basis of CIC-3 transporter inhibition by TMEM9 and PI(3,5)P2

Corresponding Author: Professor Richard Hite

Version 0:

Decision Letter:

27th Nov 2024

Dear Dr. Hite,

Thank you again for submitting your manuscript "Structural basis of CIC-3 inhibition by TMEM9 and PI(3,5)P2". I am sorry for the delay. I'm writing to let you know that we have decided to send your manuscript for peer review.

I am re-opening the manuscript submission link for you to resubmit your manuscript with all the associated files needed for the peer review process directly to our system, at your convenience. Please see below for details regarding the required materials. Please follow the link at the bottom of this email to upload the documents.

We want to ensure that the methods and statistics reporting in our papers are of the highest quality. To that end, we ask authors to fill out a Reporting Summary that collects information on experimental design and reagents, as well as an editorial Policy Checklist, which confirms compliance with our editorial policies, including the declaration of Competing Interests. If your paper includes ChIP-seq, flow cytometry or MRI data, we ask you take special care to complete those sections of the Reporting Summary as this data will aid greatly in the review of your manuscript.

These documents can be found by following the links below:

Reporting Summary:
<https://www.nature.com/documents/nr-reporting-summary.pdf>

Editorial Policy Checklist: <https://www.nature.com/documents/nr-editorial-policy-checklist.pdf>

Please be aware of our guidelines on digital image standards.

IMPORTANT

In order for us to proceed with the peer review process of your manuscript, we require you to provide accession numbers and reviewer tokens to access sequencing data sets. Please add this information to your manuscript file.

Please note we require official wwPDB validation reports for newly described atomic structures, as noted in the policy checklist. We also request that authors provide cryo-EM maps, half-maps and models, to help the reviewers in assessing the work. We recommend the use of figshare integration into our systems, which allows for provision of anonymous access links for the referees (<https://www.springernature.com/gp/authors/research-data/figshare-integration>). Alternatively, please upload .zip folders directly with the submission. To ensure the ease of reviewer access to the data, please specify in the Data Availability section, where the files can be found (provide a figshare link or direct the reader to the manuscript files).

Additionally, I would like to kindly request that you provide the code used to analyse the data to the reviewers, if used. In order for the reviewers to evaluate the work adequately they must be able to test the software/review the code themselves. If you have not yet provided the software, we therefore request that you provide a single compressed zip file containing the software with a readme.txt file or other user manual containing complete instructions for installing and running the software. If appropriate, please also provide example data and expected output. Sufficient material should be provided for referees to directly test the performance of the software/algorithm. If the software and materials are small enough to fit in a single compressed zip file less than 6MB in size, you may email this file directly to me. If the zip file is between 6 MB and 200 MB you may upload it to our file transfer site. If necessary, a second zip file up to 200 MB in size can be used to supply the example data. Please let me know if you need to use this option and I'll send you further details. Alternatively, you can also

upload the code to GitHub and provide us with the link.

Please also fill out and return to me the code and software submission checklist that will be made available to editors and reviewers during manuscript assessment. Please note that this form is a dynamic 'smart pdf' and must therefore be downloaded and completed in Adobe Reader, instead of opening it in a web browser.

<https://www.nature.com/documents/nr-software-policy.pdf>

Please use the link below to submit the files. **Please also remember to move forward all other files associated with this version of the paper.**

Link Redacted

Sincerely,

Melina

Melina Casadio, PhD
Consulting Editor, Nature Structural & Molecular Biology
Senior Editor, Nature Cell Biology
ORCID ID: <https://orcid.org/0000-0003-2389-2243>

Version 1:

Decision Letter:

8th Jan 2025

Dear Professor Hite,

Thank you again for submitting your manuscript "Structural basis of CIC-3 inhibition by TMEM9 and PI(3,5)P2". I apologize for the delay in responding, which resulted from the difficulty in obtaining referee reports. Nevertheless, we now have comments (below) from the 3 reviewers who evaluated your paper. In light of those reports, we remain interested in your study and would like to see your response to the comments of the referees, in the form of a revised manuscript.

You will see that the reviewers found the findings of interest but felt that additional work is needed to support the conclusions, in particular when it comes to the functional conclusions. To guide the scope of the revisions, we list below a prioritized set of referee points that should be addressed in the revision, which we hope will be helpful to you. Please be sure to address/respond to all concerns of the referees in full in a point-by-point response and highlight all changes in the revised manuscript text file.

-- Revs#1-2 provided detailed suggestions to extend the functional studies and ensure the mechanism of pore blocking and role of PI(3,5)P2 binding are sufficiently supported in the study (Rev#1 points #1-2 and Rev#2 points 1-2-3-4). Rev#3 also made a suggestion along these lines ("At several places..." paragraph). Each manuscript must stand on its own merits for publication in NSMB; hence, this work cannot rely on another study to support its own conclusions. The dataset in this present manuscript must be bolstered to ensure the model is convincing functionally and mechanistically to experts in this area based on data included in this submission. Thus, if publication at NSMB is to be further pursued, we would editorially expect that the aforementioned recommendations by the reviewers are followed to generate support for the functional insight and structural studies.

- Rev#3 made suggestions to refine the structural analyses, and their points should be addressed in full.

- Please also address the other reviewers' requests to strengthen the current dataset, provide additional details and discussion or other text edits, and address any minor and technical points.

We are committed to providing a fair and constructive peer-review process. Please do not hesitate to contact us if there are

specific requests from the reviewers that you believe are technically impossible or unlikely to yield a meaningful outcome or if you have any other questions or concerns.

We appreciate the requested revisions are extensive. Our standard revision period is about six months; if you cannot send it within this time, please let us know. We will be happy to consider your revision as long as nothing similar has been accepted for publication at NSMB or published elsewhere. Should your manuscript be substantially delayed without notifying us in advance and your article is eventually published, the received date would be that of the revised, not the original, version.

Reporting Summary:

EXTENDED DATA FIGURES

****If there are additional or modified structures presented in the final revision, please submit the corresponding PDB validation reports.****

Please note that all key data shown in the main figures as cropped gels or blots should be presented in uncropped form, with molecular weight markers. These data can be aggregated into a single supplementary figure. While these data can be displayed in a relatively informal style, they must refer back to the relevant figures. These data should be submitted with the last revision, prior to acceptance, but you may want to start putting it together at this point.

We require deposition of coordinates (and, in the case of crystal structures, structure factors) into the Protein Data Bank with the designation of immediate release upon publication (HPUB). Electron microscopy-derived density maps and coordinate data must be deposited in EMDB and released upon publication. Deposition and immediate release of NMR chemical shift assignments are highly encouraged. Deposition of deep sequencing and microarray data is mandatory, and the datasets must be released prior to or upon publication. To avoid delays in publication, dataset accession numbers must be supplied with the final accepted manuscript and appropriate release dates must be indicated at the galley proof stage. Please find the complete NRG policies on data availability at <http://www.nature.com/authors/policies/availability.html>.

Nature Structural & Molecular Biology is committed to improving transparency in authorship. As part of our efforts in this direction, we are now requesting that all authors identified as 'corresponding author' on published papers create and link their Open Researcher and Contributor Identifier (ORCID) with their account on the Manuscript Tracking System (MTS), prior to acceptance. This applies to primary research papers only. ORCID helps the scientific community achieve unambiguous

attribution of all scholarly contributions. You can create and link your ORCID from the home page of the MTS by clicking on 'Modify my Springer Nature account'. For more information please visit www.springernature.com/orcid.

Link Redacted

Sincerely,

Melina

Melina Casadio, PhD
Consulting Editor, Nature Structural & Molecular Biology
Senior Editor, Nature Cell Biology
ORCID ID: <https://orcid.org/0000-0003-2389-2243>

Referee expertise:

Referee #1: CLC channels

Referee #2: structural biology, channels

Referee #3: structural biology, channels

Reviewers' Comments:

Reviewer #1 (Remarks to the Author):

The study provides cryo-EM structures of CIC-3 in complex with TMEM9, a protein shown previously to inhibit CLC currents. The high-resolution view provides important molecular insight into the subunit interactions and also reveals an unanticipated PI(3,5)P2 binding site. However, it is disappointing that the paper does little to leverage the structure to formulate and test hypotheses through functional assays.

Points for Improvement:

Major:

1. C-terminal blocking mechanism: The structure leads to hypotheses that the TMEM9 C-terminal domain acts as a reversible pore blocker. The pore-blocking model is consistent with mutagenesis done in the companion paper based on sequence gazing. The manuscript would be more compelling if new predictions based on the structure were tested in functional experiments to improve mechanistic insights. Importantly, there is currently no data to support that the blocking effect is reversible.

2. Functional Validation of PI(3,5)P2 Binding.

- a. The role of PI(3,5)P2 in CIC-3/T9A interaction is inferred using apilimod, a PIKfyve inhibitor. More specific validation is needed with mutations that disrupt PI(3,5)P2 binding or the CIC-3/T9A interaction to rule out off-target effects of apilimod.
- b. Only representative images are shown. Summary data should be presented

3. Methods Section Rigor: The methods section lacks sufficient detail to allow reproducibility.

Minor:

1. Generalization to CLC4 and CLC5. The applicability of findings to CIC-4 and CIC-5 is not adequately tested beyond sequence alignment. Notably, CIC-4 and CIC-5 show only partial association with T9A and T9B unlike CLC-3's interaction with T9A and T9B, raising questions about mechanistic conservation (Extended Data Figure 1). Either wording should be toned down or experimental validation, such as vacuolization assays or structural data for CIC-4 and CIC-5, is needed to support claims about broader relevance.

2. Figure 3b shows two residues labeled as 176—phenylalanine (F) and valine (V).

3. Line 144 – There is no evidence presented that T9A is an "obligatory" beta subunit.

4. Line 193 should list 294Asn instead of 298Asn, as indicated in Figure 4a.

4. Line 207 lists the td mutation as E339A, but the methods section refers to it as E281A. These should be made consistent.

Reviewer #2 (Remarks to the Author):

In this study, the authors present cryo-EM structures of the human CIC-3 exchanger in the apo state and in complex with its accessory subunit TMEM9A. Based on their structures, the authors found that TMEM9A inhibits CIC-3 by blocking the cytosolic entrance to the Cl⁻ permeation pathway. They also showed that the phospholipid PI(3,5)P2 stabilizes the interaction between TMEM9A and CIC-3. These structures are important advances in the field of CIC proteins, as they offer key insights regarding the working mechanisms of CIC-3 and its auxiliary subunits. The data of cryo-EM maps are of high quality, with appropriate use of statistics.

However, there are several concerns to be addressed:

Major concerns:

1. One of the key findings in this study is that T9A inhibited CIC-3 as a pore blocker. To support this mechanism, the authors stated that "mutations in either T9A or CIC-3 that disrupt the interface between the T9A CD and CIC-3 diminish the ability of T9A to inhibit CIC-3" as reported in their accompanying manuscript. However, in this accompanying manuscript, I only find patch-clamp recordings of T9A mutants with CIC-5 (Fig. 2, Extended Data Fig. 5, Extended Data Fig. 9), but not with CIC-3. The authors should either clearly state the figures in the accompanying manuscript showing functional studies on T9A or T9B mutations with CIC-3, or perform the functional recordings and show the results in this manuscript.
2. If the C terminus of T9A blocks the ion permeation pathway in CIC-3 as the authors show, they should synthesize the peptide of the C terminus of T9A and apply this peptide in patch-clamp recordings to show that the C terminus peptide can directly inhibit CIC-3 in the absence of T9A or T9B co-expression.
3. Molecular dynamics simulations should be performed to test the stability of T9A binding and PIP2 binding. More importantly, MD will also test whether the C terminus of T9A can stably bind to block the ion permeation port.
4. Point mutations around the PIP2 binding sites should be introduced and tested with patch-clamp recordings. For instance, mutations to R163 in T9A or R254 in CIC-3 are expected to largely perturbate T9A inhibition.

Minor concerns:

1. A panel showing the structural alignment of the human apo state and the T9A bound state is needed.
2. In the Abstract "whose mutation are associated with" should be corrected to "whose mutations are associated with."
3. In main text (Line 165) "acts a pore blocker" should be corrected to "acts as a pore blocker."

Reviewer #3 (Remarks to the Author):

In this highly interesting manuscript, the authors report the structural basis for the interaction of the endosomal Cl⁻/H⁺ exchanger CIC3 with its β -subunits TMEM9A/B and the lipid PI(3,5)P2. As common interaction partners of a subgroup of the CLC family, encompassing CLC3-5, which all play a role in the maturation of endosomal compartments, TMEM9A and B are critical for the targeting and regulation of CLC activity. The work presents cryo-EM structures of high quality in combination with cellular assays to show how both interaction partners might down-regulate CIC3 activity as described in an accompanying and still unpublished study. The manuscript is generally well written and the claims are for most parts conclusive. I thus see the work as strong candidate for publication in Nature Structural and Molecular Biology.

I have few remarks that should be addressed:

- The authors frequently refer to Cl⁻ permeation pathway which is a proper term for an ion channel and not commonly used for a secondary transporter. Arguably CLC transporter show channel-like features but the authors could better explain why they use this nomenclature.
- Similarly, the term pore blocker in Line 169 needs to be explained in this context.
- At several places the authors mention the inhibition of CLC3 function by TMEM9A and refer to a complementary manuscript that is still unpublished. It could be helpful if the authors would add a simple experiment showing this inhibition in this manuscript even if it is somewhat redundant to the accompanying manuscript. However, I do not consider the inclusion of such data essential.
- In line 175, the authors suggest that TMEM9A would have to dissociate to relief the inhibitory effect. Are there alternative regulatory processes conceivable that would not require the complete dissociation of the protein?
- Do the authors have any biochemical evidence that PI(3,5)P2 was co-purified with their sample?
- When describing the structure of CLC3, I believe the authors refer to the dataset #1 in Extended Table 1. For this dataset a magnification of 29,000 was noted while the magnification of all other datasets was defined as 165,000. Is the first number correct?
- The authors refer to an ATP-binding site located in the CTD, which they claim was not occupied in their structure. I wonder how strong the evidence for ATP binding to CLC3 is. Is the site conserved compared to CLC5? The structural features of this region could be better documented. It would also be interesting to know whether the authors have added any ATP to their

sample before vitrification.

- The density for the bound PI(3,5)P2 lipid and the surrounding residues could be better documented.
- Extended Data Fig. 1 shows size exclusion profiles of different CLC paralogs in complex with TMEM9. As far as I understand, the fluorescence of the tagged CLC protein was monitored with peak shifts indicating complex formation. I wonder whether the authors have also recorded the fluorescence of the Venus-tagged TMEM9 protein, where the peak shift should be much more pronounced.

Minor:

- It would be helpful if the panels in Extended Data Fig. 3 would contain the name of the displayed protein.
- The panels in Extended Data Fig. 2 are very small and barely discernible.
- Line 75 correction: Distinct Cl⁻ permeation pathways
- Line 118 correction: Highly conserved in CIC-3,4,5.
- The described interaction of the side chain of Arg 172 with the backbone oxygens of Lys 230 and Val 231 cannot be appreciated in Figure 3b.
- The PI(3,5)P2 molecule is barely visible in Fig. 2. The authors might consider to increase the thickness of the stick model.
- The interactions of the PI(3,5)P2 with the protein could be better illustrated. Particularly interactions with the protein backbone cannot be appreciated with in the ribbon model of the protein. Closeups with superimposed density would allow a better appreciation of the data supporting the described conclusions.
- The panels showing immunofluorescence data in Fig. 4 b-e are very small and sometimes difficult to appreciate.
- The interaction between the Asp 87 sidechain and the backbone carbonyl of Arg 254 displayed in Fig. 4f would require a protonated Asp sidechain. I wonder how well this interaction is defined in the density and whether there is any indication for an increase of the pKa of the sidechain.
- The interaction between the backbone carbonyl of Ile 252 is drawn to the C α atom.

Version 2:

Decision Letter:

23rd Apr 2025

Dear Dr. Hite,

Thank you again for submitting your revised manuscript "Structural basis of CIC-3 inhibition by TMEM9 and PI(3,5)P2". The revision has now been assessed by the original reviewers.

You will see that all reviewers find that the conclusions are now stronger. Revs#2-3 in particular are supportive of publication. Rev#1 requests clarifications and additional details to fully assess the revisions.

While we limit all our manuscripts to a single round of major experimental revision to limit the overall time spent in peer review, the changes requested at this stage are relatively minor. We therefore are open to a final round of minor revision and hope you can submit your revision within 6 weeks. Please let me know if you expect this to be an issue. We ask that you please address Rev#1's points in full as they pertain to the revision. Please note that we might consult Rev#1 again at resubmission.

In addition, we ask that you please note the changes that will be required to meet our formatting requirements for publication:

1- We can accommodate up to 8 display items (= Figures and main Tables) in the main article and up to 10 Extended Data figures, which will be integrated into the full-text HTML version of your paper and will be appended to the online PDF.

Each Extended Data item must be cited in order in the main text. Each Figure, Table and Extended Data figure must fit easily within an A4 page (210 x 297 mm). Please ensure that the number and size of your Figures, Tables and Extended Data figures fulfil these requirements to avoid any delay in the acceptance of your article.

2- Please make the cryo-EM table Table 1 (not ED 1).

3- Please make sure figures are all limited to 1 page max. Currently ED2 spans two pages. All panels must fit on one page,

roughly A4 in size, portrait orientation, for all main and ED figures. Please also be sure to respect our format limits (8 main display items and 10 ED figures max).

4- Please ensure your main manuscript file includes the following sections, in this order:

Title
Author list
Affiliations
Abstract
Main text
Acknowledgements
Author Contributions Statement
Competing Interests Statement
Tables
Figure Legends/Captions (for main text figures)
References
Methods
Data Availability
Code Availability (if relevant)
Methods-only references

5-Please refrain from using words such as new/novel/first, when referring to the scientific findings.

6- Our editorial guidelines stipulate that the main text of articles should not exceed 4500 words (excluding the Methods section and references section).

7- All micrographs should have scale bars, including magnifications. For instance, scale bars seem to be currently missing on magnifications in figure 4.

We expect to see your revised manuscript within 6 weeks. If you cannot send it within this time, please contact us to discuss an extension; we would still consider your revision, provided that no similar work has been accepted for publication at NSMB or published elsewhere.

Reporting Summary:

EXTENDED DATA FIGURES

Please note that all key data shown in the main figures as cropped gels or blots should be presented in uncropped form, with molecular weight markers. These data can be aggregated into a single supplementary figure item. While these data can be

displayed in a relatively informal style, they must refer back to the relevant figures. These data should be submitted with the final revision, as source data, prior to acceptance, but you may want to start putting it together at this point.

Data availability: this journal strongly supports public availability of data. All data used in accepted papers should be available via a public data repository, or alternatively, as Supplementary Information. If data can only be shared on request, please explain why in your Data Availability Statement, and also in the correspondence with your editor. Please note that for some data types, deposition in a public repository is mandatory - more information on our data deposition policies and available repositories can be found below:

<https://www.nature.com/nature-research/editorial-policies/reporting-standards#availability-of-data>

Link Redacted

Sincerely,

Melina Casadio, PhD
Locum Chief Editor, Nature Structural & Molecular Biology
ORCID ID: <https://orcid.org/0000-0003-2389-2243>

Reviewers' Comments:

Reviewer #1 (Remarks to the Author):

The authors have conducted experiments to investigate the interaction between T9A CD and CLC-3, as identified from their cryo-EM structures. By introducing mutations at the interface and assessing their effects using a vacuolization assay, they demonstrate that these mutations reduce T9A's efficacy in inhibiting CLC-3 activity. Additionally, the authors explore the role of residues involved in PI(3,5)P₂ coordination using the same assay. The results appear to support the proposed interactions and thus address the previous criticism. However, full evaluation of the data is not possible, as a few key points in the approach require clarification and transparency.

1. Methods: The 'Detection of large vacuoles by microscopy' section in the Methods lacks sufficient detail for reproducibility. Please specify the ImageJ analysis workflow, including any thresholding methods, segmentation criteria for vacuoles, and any pre-processing steps applied. The method of calculating values presented in Figures 3k, 4i, and Extended Data 9b should be spelled out in detail. Additionally, clarification is needed on whether biological or technical replicates were performed.

2. Figure labels and legends: The y-axis labels in Figure 3k and Extended Data Figure 9b are ambiguous and confusing. It

appears to be a distribution of cells based on vacuole area in those cells. But it is unclear how the binning is done to get the distribution here. Is the 'vacuole area' total area occupied by all vacuoles in the cell? Or of a single 'typical' vacuole in the cell? 'Typical' assumes uniform vacuole distribution size in a cell – is that assumption justified? The y-axis label should be revised and given a precise and complete definition in the legend.

3. Include summary tables: It is difficult to read the data and error bars in the plots. While the plots provide a nice qualitative summary of the data (once points #1 and 2 are addressed), a summary table showing the values for the data presented in Figures 3k, 4i and Extended Data Figure 9b is essential for transparency and data interpretation.

Reviewer #2 (Remarks to the Author):

In the revised manuscript, the authors have sufficiently addressed all my concerns.

Reviewer #3 (Remarks to the Author):

The authors have addressed all comments in a satisfactory manner. The manuscript is improved and ready for publication.

Version 3:

Decision Letter:

Our ref: NSMB-A50162C

16th May 2025

Dear Dr. Hite,

Thank you for submitting your revised manuscript "Structural basis of CIC-3 inhibition by TMEM9 and PI(3,5)P2" (NSMB-A50162C). It has now been seen by the original referee Rev#1 and their comments are below. The reviewer finds that the paper has improved in revision, and therefore we'll be happy in principle to publish it in Nature Structural & Molecular Biology, pending minor revisions to satisfy the referee's final requests and to comply with our editorial and formatting guidelines.

We are now performing detailed checks on your paper and will send you a checklist detailing our editorial and formatting requirements in about 1-2 weeks. Please do not upload the final materials and make any revisions until you receive this additional information from us.

Thank you again for your interest in Nature Structural & Molecular Biology. Please do not hesitate to contact me if you have any questions.

Sincerely,

Melina Casadio, PhD
Locum Chief Editor, Nature Structural & Molecular Biology
ORCID ID: <https://orcid.org/0000-0003-2389-2243>

Reviewer #1 (Remarks to the Author):

The authors have effectively responded to the critique and made the necessary revisions to the manuscript.

There is just one minor typo to be fixed:

> In Figure 3e, the units should read 1.0–1.5 μm^2 and 1.5 μm^2 — the superscript 2 is currently missing.

We thank the reviewers for their critical assessment of our work and their helpful suggestions. In order to address concerns raised by the reviewers, we have expanded our investigation into the regulation of endosomal CLC transporters by T9A and PI(3,5)P₂. We have (1) used mutagenesis to identify residues at the interface between CIC-3 and the T9A CD that are critical for inhibition, (2) assessed the effect of a structurally-distinct PIKfyve inhibitor on the activity of CIC-3 in endosomes, (3) demonstrated that the activity of CIC-5 is regulated by PI(3,5)P₂ in endosomes, and (4) mutated residues in CIC-3 and T9A that participate in the coordination of PI(3,5)P₂. Collectively, these results support our proposal that the T9A CD occludes the Cl⁻ ion pathway when bound to CIC-3 and the equilibrium between bound and unbound states is influenced by the presence of PI(3,5)P₂. Sections of the text describing these new results are highlighted with a blur bar on the left side. A point-by-point response to the reviewers' comments is below.

Reviewer #1 (Remarks to the Author):

The study provides cryo-EM structures of CIC-3 in complex with TMEM9, a protein shown previously to inhibit CLC currents. The high-resolution view provides important molecular insight into the subunit interactions and also reveals an unanticipated PI(3,5)P₂ binding site. However, it is disappointing that the paper does little to leverage the structure to formulate and test hypotheses through functional assays.

We thank the reviewer for their critical assessment of our work and their helpful suggestions that have improved our study.

Points for Improvement:

Major:

1. C-terminal blocking mechanism: The structure leads to hypotheses that the TMEM9 C-terminal domain acts as a reversible pore blocker. The pore-blocking model is consistent with mutagenesis done in the companion paper based on sequence gazing. The manuscript would be more compelling if new predictions based on the structure were tested in functional experiments to improve mechanistic insights. Importantly, there is currently no data to support that the blocking effect is reversible.

As the reviewer noted, in the companion report we describe mutations in the cytosolic domains of T9A and T9B that diminish their inhibitory effects on CIC-3. To further interrogate the role of the interface between the T9A CD and CIC-3, we mutated residues that comprise this interface and assayed their effect on CIC-3 mediated vacuolization. In new data added to the revised Figure 3, we show that the T9A^{W164A} and T9A^{R172A} mutants are less effective than wild-type T9A at inhibiting CIC-3 activity. ¹⁶⁴Trp forms a π - π interaction with W260 of CIC-3 and a hydrophobic interaction with F232 of CIC-3. R172 of T9A interacts with the backbone oxygen atoms of K230 and V231 of CIC-3. We also mutated W260 of CIC-3, finding that the CIC-3^{W260A} mutant could not be completely inhibited by T9A. Together, these analyses highlight critical interactions that stabilize the interactions between CIC-3 and the T9A cytosolic domain.

In our initial submission, we proposed that the blocking effect of the T9A CD on CIC-3 is reversible. This proposal was based on the accessibility of the cytosolic domains of T9A/B and the bound PI(3,5)P₂ to enzymatic modification. In our companion report, we note previous proteomic analyses have identified residues in the cytosolic domains of T9A and T9B can be phosphorylated or ubiquitylated (www.phosphosite.org). These residues are unlikely to be accessible for modification when the cytosolic domains are bound to CLCs, suggesting that the association between CIC-3 and the T9A CD is dynamic. However, our approaches do not inform on the kinetics of the association and dissociation of the T9A CD. We have therefore removed the terms "reversible" and "reversibly" from our manuscript. Future experiments outside the scope of the current study will be required to investigate the timescales at which the blocking effect occurs.

2. Functional Validation of PI(3,5)P₂ Binding.

a. The role of PI(3,5)P₂ in CIC-3/T9A interaction is inferred using apilimod, a PIKfyve inhibitor. More specific validation is needed with mutations that disrupt PI(3,5)P₂ binding or the CIC-3/T9A interaction to rule out off-target effects of apilimod.

We thank the reviewer for suggesting approaches that would strengthen our analysis of the role of PI(3,5)P₂ in contributing to the regulation of CIC-3/T9A. In our initial submission, we found that apilimod treatment had no effect on cells co-expressing transport-deficient CLC-3^{td} mutant and T9A, suggesting that the apilimod's effect on vacuolization requires the transport activity of CIC-3 (presented in the revised Figure 4). To further investigate the role of potential off-target effects, we have assessed the effect of YM-201636, a PIKfyve inhibitor that is structurally distinct from apilimod and other WX8-family PIKfyve inhibitors (PMID: 30806145), on CIC-3 activity using the venus-CIC-3 vacuolization assay. In cells that co-express CIC-3 and T9A, we find that YM-201636 treatment promotes vacuolization in a manner that is similar to that observed for apilimod. The similar effects of YM-201636 and apilimod on vacuolization, despite their distinct chemical structures, strongly indicates that they alter vacuolization by modulating PI(3,5)P₂ abundance rather through off-target effects. These new data are presented in the new Extended Data Figure 8.

We also employed the venus-CIC-3 vacuolization assay to probe the role of residues involved in the coordination of PI(3,5)P₂. We first examined ²⁵⁴Arg and ³¹⁰Lys of CIC-3, which interact with the phosphate at the 3 position of PI(3,5)P₂. Whereas vacuolization induced by wild-type CIC-3 is suppressed by co-expression with T9A, T9A cannot suppress the vacuolization induced by the CIC-3^{R254A} and CIC-3^{K310A} mutants. We also mutated ¹⁶³Arg of T9A, which interacts with the phosphate at the 5 position of PI(3,5)P₂. Although the T9A^{R163A} largely suppressed vacuolization induced by CIC-3, the T9A^{R163L} variant, which we identified as an endogenously occurring variant in a cancer data base, led to an increase in vacuolization. These new results are presented in the revised Extended Data Figure 9. Together, these results show that chemical depletion of PI(3,5)P₂ or genetic disruption of the PI(3,5)₂ binding site diminishes the inhibitory effects of T9A on endosomal CIC-3.

b. Only representative images are shown. Summary data should be presented

Summary data are presented in the revised versions of Figures 3, 4 and Extended Data Figure 9.

3. Methods Section Rigor: The methods section lacks sufficient detail to allow reproducibility.

The methods section has been updated to provided sufficient detail to allow reproducibility.

Minor:

1. Generalization to CLC4 and CLC5. The applicability of findings to CIC-4 and CIC-5 is not adequately tested beyond sequence alignment. Notably, CIC-4 and CIC-5 show only partial association with T9A and T9B unlike CLC-3's interaction with T9A and T9B, raising questions about mechanistic conservation (Extended Data Figure 1). Either wording should be toned down or experimental validation, such as vacuolization assays or structural data for CIC-4 and CIC-5, is needed to support claims about broader relevance.

In Figure 1 of our companion report (*in press* at Nature Communication and attached as a PDF), we show that CIC-3, CIC-4 and CIC-5 interact with T9A and T9B in mouse brain tissue and can be co-immunoprecipitated from tissue culture cells. We further show in Figure 6 of the companion report that CIC-4 and CIC-5 are inhibited by T9A using the vacuolization assay. The FSEC analyses that we present in Extended Data Figure 1 show that CIC-3 forms a complex with T9A in detergent that remains intact for several hours and was thus deemed suitable for cryo-EM analysis. Our analysis indicated that CIC-4 and CIC-5 also formed complexes with T9A, but that the assemblies were sub-stoichiometric. Notably, the FSEC analyses also revealed broader peaks for CIC-4 and CIC-5 when expressed alone, indicating that CIC-4 and CIC-5 may be less biochemically stable in the conditions used for purification. To directly assess if the CLC constructs used for FSEC can interact with T9A, we performed co-immunoprecipitation assays, finding that mCerulean-tagged CIC-3, CIC-4 and CIC-5 immunoprecipitate strep-tagged T9A to varying degrees, confirming that CIC-3, CIC-4 and CIC-5 can interact with T9A. These new data are presented in the revised Extended Data Figure 1.

To assess if PI(3,5)P₂ plays a conserved role in the regulation of the CIC-3 to CIC-5 clade of CLCs, we co-expressed CIC-5 and T9A in HeLa cells lacking CIC-7. Vacuolization was observed in cells treated with apilimod, but not in the untreated cells, indicating that CIC-5 is inhibited by T9A in a PI(3,5)P₂-dependent manner. These new data are presented in the revised Figure 4. Thus, the inhibition of CIC-3 and CIC-5 by T9A requires PI(3,5)P₂. Although we do not directly assess if

the inhibition of CIC-4 by T9A also requires a PI(3,5)P₂, we think it highly likely as the residues that coordinate PI(3,5)P₂ are conserved across the clade.

2. Figure 3b shows two residues labeled as 176—phenylalanine (F) and valine (V).

Corrected

3. Line 144 – There is no evidence presented that T9A is an “obligatory” beta subunit.

We have cited our companion report on line 149 in which it was demonstrated that either T9A or T9B are obligatory beta subunits of CIC-3, CIC-4 and CIC-5.

4. Line 193 should list 294Asn instead of 298Asn, as indicated in Figure 4a.

Corrected

4. Line 207 lists the td mutation as E339A, but the methods section refers to it as E281A. These should be made consistent.

We thank the reviewer for noting this inconsistency. E339 corresponds to the long isoform that we used for structural analysis. E281 corresponds to the short CIC-3a isoform used for the vacuolization assay. We have now clarified in the methods that we used the E281A mutant of the short isoform that corresponds to the E339A mutant for the long isoform used for structural analysis.

Reviewer #2 (Remarks to the Author):

In this study, the authors present cryo-EM structures of the human CIC-3 exchanger in the apo state and in complex with its accessory subunit TMEM9A. Based on their structures, the authors found that TMEM9A inhibits CIC-3 by blocking the cytosolic entrance to the Cl⁻ permeation pathway. They also showed that the phospholipid PI(3,5)P₂ stabilizes the interaction between TMEM9A and CIC-3. These structures are important advances in the field of CIC proteins, as they offer key insights regarding the working mechanisms of CIC-3 and its auxiliary subunits. The data of cryo-EM maps are of high quality, with appropriate use of statistics.

We thank the reviewer for their assessment of our work and their suggestions for ways to improve the manuscript.

However, there are several concerns to be addressed:

Major concerns:

1. One of the key findings in this study is that T9A inhibited CIC-3 as a pore blocker. To support this mechanism, the authors stated that “mutations in either T9A or CIC-3 that disrupt the interface between the T9A CD and CIC-3 diminish the ability of T9A to inhibit CIC-3” as reported in their accompanying manuscript. However, in this accompanying manuscript, I only find patch-clamp recordings of T9A mutants with CIC-5 (Fig. 2, Extended Data Fig. 5, Extended Data Fig. 9), but not with CIC-3. The authors should either clearly state the figures in the accompanying manuscript showing functional studies on T9A or T9B mutations with CIC-3, or perform the functional recordings and show the results in this manuscript.

We thank the reviewer for the suggestion. To investigate the regulation of CIC-3 by T9A, we used the vacuolization assay developed in our companion report (*in press* at Nature Communication and attached as a PDF). In Figures 4-6 of the companion report, we show that this assay monitors the activity of CIC-3, CIC-4 and CIC-5 in endosomes and that the activity of these transporters can be inhibited by T9A and T9B. In this report, the vacuolization assay was used to assess the effect of mutations in T9A on CIC-3 activity because we found two distinct mechanisms in our companion report through which T9A and T9B suppress the activity of CIC-5. First, we show in Figure 2 of the companion report that co-expression of T9A or T9B greatly reduces the plasma membrane expression of CIC-5. Second, we further show that T9A and T9B also directly inhibit the activity of CIC-5. Since vacuolization assesses the activity of CIC-3 in its native endosomal environment, we can evaluate the effects of mutations on CIC-3 activity. In the revised version of Figure 3, we identify residues at the interface between CIC-3 and the T9A CD that are necessary for T9A to inhibit CIC-3, including W260 of CIC-3 and W164 and R172 of T9A.

2. If the C terminus of T9A blocks the ion permeation pathway in CIC-3 as the authors show, they should synthesize the peptide of the C terminus of T9A and apply this peptide in patch-clamp recordings to show that the C terminus peptide can directly inhibit CIC-3 in the absence of T9A or T9B co-expression.

We thank the reviewer for this suggestion. However, the currents generated by CIC-3 are too small to detect using excised patch clamp approaches that would allow us to manipulate the cytosolic buffer composition. In Figure 2 of our companion report, we present two-electrode voltage clamp recordings of plasma membrane localized CIC-5. In this configuration, the currents are larger, but the intact plasma membrane prevents manipulation of the cytosolic buffer composition.

Binding assays presented in Figure 1 of the companion report and the structural analyses presented here demonstrate that T9 binding to CIC-3 is governed by interactions in the transmembrane and luminal domains. Our finding that disruption of the PI(3,5)P₂-mediated interaction diminishes the inhibitory effect of T9A indicates that the cytosolic domain only weakly interacts with CIC-3. It is therefore highly unlikely that the cytosolic domain by itself would be sufficient to inhibit ion transport.

3. Molecular dynamics simulations should be performed to test the stability of T9A binding and PIP₂ binding. More importantly, MD will also test whether the C terminus of T9A can stably bind to block the ion permeation port.

We thank the reviewer for the suggestion. However, we find it unlikely that molecular dynamics simulations will provide a significant gain in understanding how T9A and PI(3,5)P₂ interact with CIC-3. We would instead preferred to use additional functional analyses and mutagenesis to validate our model. During revision, we have demonstrated that the inhibition of CIC-5 by T9A depends on the presence of PI(3,5)P₂, indicating that our proposed mechanism is conserved across the CIC-3 to CIC-5 clade (new data is presented in revised versions of Figure 4 and Extended Data Figure 8). We have also demonstrated that mutation of residues at the interface between CIC-3 and T9A lead to disinhibition of CIC-3 activity, as does mutation of residues in CIC-3 and T9A that coordinate the PI(3,5)P₂ headgroup (new data is presented in revised versions of Figure 3 and and Extended Data Figure 9). Together, these experiments provide strong support for our model.

4. Point mutations around the PIP2 binding sites should be introduced and tested with patch-clamp recordings. For instance, mutations to R163 in T9A or R254 in CIC-3 are expected to largely perturbate T9A inhibition.

We thank the reviewer for their suggestion. A previous report (ref 28 in our manuscript) demonstrated that T9B can diminish CIC-3 currents recorded from oocytes. In Figure 2 of our companion report (the revised version which is attached to this submission and in press at Nature Communications), we show that T9A and T9B can diminish CIC-5 currents in oocytes through two mechanisms. In addition to directly inhibiting ion transport, T9A and T9B also greatly reduce the plasma membrane expression of CIC-5. To eliminate the confounding effects of membrane trafficking, we have investigated the activity of CIC-3 in endosomal membranes by monitoring the appearance of enlarged, acidified vacuoles. Using this CIC-3 induced vacuolization assay, we have demonstrated that several of the residues that participate in the coordination of the PI(3,5)P₂ headgroup, including T9A-R163, CIC-3-R254 and CIC-3-K310, contribute to the regulation of CIC-3 by T9A. These new data are presented in the revised version of Extended Data Figure 9.

Minor concerns:

1. A panel showing the structural alignment of the human apo state and the T9A bound state is needed.

We present structural alignments and all-atom RMSD calculations for CIC-3 alone (grey) with CIC-3/T9A (blue – CIC-3/yellow – T9A), CIC-3 alone (grey) with CIC-3/noT9A (blue – CIC-3) and CIC-3 alone (grey) with two classes of CIC-3 in complex with T9A in which the T9A protomers are only partially ordered (blue – CIC-3/yellow – T9A) in Extended Data Figure 4b. We have revised the labeling and figure legend to clarify that panel b shows these structural comparisons.

2. In the Abstract "whose mutation are associated with" should be corrected to "whose mutations are associated with."

Corrected

3. In main text (Line 165) "acts a pore blocker" should be corrected to "acts as a pore blocker."

Corrected

Reviewer #3 (Remarks to the Author):

In this highly interesting manuscript, the authors report the structural basis for the interaction of the endosomal Cl⁻/H⁺ exchanger CLC3 with its β -subunits TMEM9A/B and the lipid PI(3,5)P₂. As common interaction partners of a subgroup of the CLC family, encompassing CLC3-5, which all play a role in the maturation of endosomal compartments, TMEM9A and B are critical for the targeting and regulation of CLC activity. The work presents cryo-EM structures of high quality in combination with cellular assays to show how both interaction partners might down-regulate CLC3 activity as described in an accompanying and still unpublished study. The manuscript is generally well written and the claims are for most parts conclusive. I thus see the work as strong candidate for publication in Nature Structural and Molecular Biology.

We thank the reviewer for their positive assessment of our work and their helpful suggestions to improve our study.

I have few remarks that should be addressed:

- The authors frequently refer to Cl⁻ permeation pathway which is a proper term for an ion channel and not commonly used for a secondary transporter. Arguably CLC transporter show channel-like features but the authors could better explain why they use this nomenclature.

We have revised the text from Cl⁻ permeation pathway to the more commonly used Cl⁻ ion pathway.

- Similarly, the term pore blocker in Line 169 needs to be explained in this context.

We have revised the text to describe the inhibition of CLC-3 by T9A as a “ball-and-chain like mechanism”, which is a less channel specific nomenclature.

- At several places the authors mention the inhibition of CLC3 function by TMEM9A and refer to a complementary manuscript that is still unpublished. It could be helpful if the authors would add a simple experiment showing this inhibition in this manuscript even if it is somewhat redundant to the accompanying manuscript. However, I do not consider the inclusion of such data essential.

We thank the reviewer for their suggestion. The companion report is in press at Nature Communications and we have attached a copy of the PDF to this submission. In order to address some of the reviewers' comments, we have used the vacuolization assay developed in the companion report to interrogate the role of specific residues at the interface between CLC-3 and the T9A CD. We first introduce this approach in the revised Figure 3, and then later in Figure 4, Extended Data Figure 8 and Extended Data Figure 9. We describe the rationale for this approach on line 180.

- In line 175, the authors suggest that TMEM9A would have to dissociate to relief the inhibitory effect. Are there alternative regulatory processes conceivable that would not require the complete dissociation of the protein?

We thank the reviewer for providing us with an opportunity to clarify our proposed mechanism. We propose that CLC-3 and T9A can exist in two functionally distinct configurations. In the inhibitory configuration, the three domains of T9A (luminal, transmembrane and cytosolic) interact with CLC-3 in the manner observed in the CLC-3/T9A structure. In this conformation, the cytosolic entrance to the Cl⁻ ion pathway is blocked by the cytosolic domain of T9A. In the disinhibited state, the cytosolic domain of T9A dissociates from CLC-3, allowing Cl⁻ ions access to the ion pathway. Notably, we think it is likely that LD and TMD remain bound to CLC-3 when the CD dissociated as binding data and our structures indicate that the CD is not required for its interaction with CLC-3. Thus, we propose that the activity of CLC-3 is determined by the binding status of the T9A cytosolic domain at the cytosolic entrance to the Cl⁻ ion pathway. This model allows modifications to the interfacial PI(3,5)P₂, to the T9A cytosolic domain, and to CLC-3 to bias the equilibrium between active and inhibited states because T9A remains associated with CLC-3 at all times. We have revised the discussion to clarify our proposed mechanism.

“In this study, we reveal how the T9A CD regulates the activity of the CLC-3 to CLC-5 clade of endosomal chloride-proton exchangers through a mechanism that appears to be conserved in T9B. When bound to cytosolic surface of CLC-3, the T9A CD physically occludes the cytosolic entrance to the CLC-3 Cl⁻ ion pathway and inhibits ion transport. The T9A CD is flexibly tethered to the TMD, which is likely always associated with CLC-3 in endosomes. Notably, T9A engagement with

CLC-3 is insufficient to fully inhibit its activity. Rather, the equilibrium of the T9A CD between the bound, inhibited state and the unbound, disinhibited state and thus the activity of CLC-3 is influenced by the endolysosomal signaling lipid, PI(3,5)P₂. PI(3,5)P₂ binds at the interface between CLC-3 and the T9A CD, stabilizing the bound, inhibited state. Regulation by PI(3,5)P₂ abundance provides a mechanism to control CLC activity in endosomes through the generation and breakdown of PI(3,5)P₂ by the kinase PIKfyve and the PtdIns(3,5)P₂ 5-phosphatase, FIG4, respectively^{41,42}. Post-translational modifications, such as phosphorylation of the T9A CD or interacting residues in CLCs²⁹, may further influence the equilibrium between bound and unbound states. These modifications would enable cells to finely tune endosomal CLC activity in response to specific stimuli.”

A plausible alternative to our proposal is that the T9A cytosolic domain may undergo a local conformational change that would enable ions to access the cytosolic entrance to the Cl⁻ ion pathway without the CD disengaging from CLC-3. Although our data does not formally exclude this alternative model, we do not observe local conformational changes of the T9A cytosolic domain in our cryo-EM analysis. Rather, we resolved several classes in our cryo-EM data in which the cytosolic domain of T9A is partially or completely disordered while the luminal and transmembrane domains remain bound to CLC-3, which would be consistent with our proposed model.

- Do the authors have any biochemical evidence that PI(3,5)P₂ was co-purified with their sample?

The assignment that the non-protein density corresponds to a PI(3,5)P₂ is based on the cryo-EM density map. The quantity of lipid co-purified with the protein is too low to be detected by TLC analysis and mass spectroscopic analyses cannot easily discriminate between PI(3,5)P₂, PI(3,4)P₂, and PI(4,5)P₂ due to their identical mass-to-charge ratios. To aid in distinguishing between these phosphatidylinositol species, we have evaluated the effect of YM-201636, another PIKfyve inhibitor on vacuolization. In cells that co-express CLC-3 and T9A, we find that YM-201636 treatment promotes vacuolization in a manner that is similar to that observed for apilimod, supporting our proposal that PI(3,5)P₂ participates in the T9A-mediated inhibition of CLC-3. These new data are presented in Extended Data Figure 8.

- When describing the structure of CLC3, I believe the authors refer to the dataset #1 in Extended Table 1. For this dataset a magnification of 29,000 was noted while the magnification of all other datasets was defined as 165,000. Is the first number correct?

The cryo-EM images of CLC-3 alone were collected on a Titan Krios equipped with a K3 detector at a nominal magnification of 29,000x, which corresponds to super-resolution pixel size of 0.413 Å. The images were Fourier cropped during analysis to yield a final pixel size of 0.826 Å. The cryo-EM images of CLC-3 in the presence of T9A were collected on a Titan Krios equipped with SelectrisX energy filter and FalconIV detector at a nominal magnification of 165,000x, which corresponds to pixel size of 0.725 Å. We have revised the methods and Extended Data Table 1 to highlight that we used microscopes with different detectors to collect the data.

- The authors refer to an ATP-binding site located in the CTD, which they claim was not occupied in their structure. I wonder how strong the evidence for ATP binding to CLC3 is. Is the site conserved compared to CLC5? The structural features of this region could be better documented. It would also be interesting to know whether the authors have added any ATP to their sample before vitrification.

A recent study investigated the adenine nucleotide binding site in mouse CLC-3 (ref 35 - PMID: 39107281). The authors found that addition of ATP prior to vitrification resulted in the appearance of a bound ATP at the interface between the cytosolic CBS domains, where ATP has been previously resolved in crystal structures of a fragment of the cytosolic domain of CLC-5 (ref 34 - PMID: 17195847). The residues comprising the ATP-binding site are highly conserved between mouse CLC-3 and CLC-5. They are also conserved in human CLC-3 but the site is unoccupied in our reconstructions calculated from images of particles vitrified in the absence of ATP. We have now clarified that no exogenous ligands were added to our samples.

- The density for the bound PI(3,5)P₂ lipid and the surrounding residues could be better documented.

We have added two additional panels to the revised version of Extended Data Figure 7 to better present the density in the vicinity of the PI(3,5)P₂. As suggested below, we removed the cartoon depiction from these panels to enable visualization of the interactions with the protein backbone.

- Extended Data Fig. 1 shows size exclusion profiles of different CLC paralogs in complex with TMEM9. As far as I understand, the fluorescence of the tagged CLC protein was monitored with peak shifts indicating complex formation. I wonder whether the authors have also recorded the fluorescence of the Venus-tagged TMEM9 protein, where the peak shift should be much more pronounced.

We appreciate the reviewer's suggestion. We did initially characterize the migration of the CLCs using mCerulean fluorescence and the TMEM9s using mVenus fluorescence. However, we found that it was not possible to detect a distinct peak for the TMEM9s. When T9A or T9B were expressed by themselves, we observed a broad profile that is consistent with a wide range of molecular weights. We hypothesized that this profile may arise from improper folding or aggregation of the TMEM9s when they are not co-expressed with CLCs but did not pursue the migration of the TMEM9s further.

Minor:

- It would be helpful if the panels in Extended Data Fig. 3 would contain the name of the displayed protein.

Revised as suggested.

- The panels in Extended Data Fig. 2 are very small and barely discernible.

We have split the workflows for the two data sets to allow the panels to improve readability.

- Line 75 correction: Distinct Cl⁻ permeation pathways

Corrected

- Line 118 correction: Highly conserved in CIC-3,4,5.

Corrected

- The described interaction of the side chain of Arg 172 with the backbone oxygens of Lys 230 and Val 231 cannot be appreciated in Figure 3b.

We have added Figure 3h to better depict this interaction.

- The PI(3,5)P₂ molecule is barely visible in Fig. 2. The authors might consider to increase the thickness of the stick model.

We thank the reviewer for their suggestion and have revised the figure accordingly.

- The interactions of the PI(3,5)P₂ with the protein could be better illustrated. Particularly interactions with the protein backbone cannot be appreciated with in the ribbon model of the protein. Closeups with superimposed density would allow a better appreciation of the data supporting the described conclusions.

As described above, we have added two additional panels to Extended Data Figure 7 to better present the interactions with the PI(3,5)P₂. We have also added Figure 4b to highlight the interactions between CIC-3/T9A and PI(3,5)P₂.

- The panels showing immunofluorescence data in Fig. 4 b-e are very small and sometimes difficult to appreciate.

We thank the reviewer for the suggestion and have revised our presentation of the imaging data.

- The interaction between the Asp 87 sidechain and the backbone carbonyl of Arg 254 displayed in Fig. 4f would require a protonated Asp sidechain. I wonder how well this interaction is defined in the density and whether there is any indication for an increase of the pKa of the sidechain.

We thank the reviewer for noting a mistake that we made when preparing the figure. The sidechain of Asp87 is 2.9 Å from the backbone nitrogen atom of Arg254, with which it would be able to make a favorable interaction. The line is shown correctly in the revised version of Extended Data Figure 10b.

- The interaction between the backbone carbonyl of Ile 252 is drawn to the C α atom.

We have corrected the dashed line in the revised version of Extended Data Figure 10.

Detailed responses to the reviewers

Reviewers' Comments:

Reviewer #1 (Remarks to the Author):

The authors have conducted experiments to investigate the interaction between T9A CD and CLC-3, as identified from their cryo-EM structures. By introducing mutations at the interface and assessing their effects using a vacuolization assay, they demonstrate that these mutations reduce T9A's efficacy in inhibiting CLC-3 activity. Additionally, the authors explore the role of residues involved in PI(3,5)P₂ coordination using the same assay. The results appear to support the proposed interactions and thus address the previous criticism. However, full evaluation of the data is not possible, as a few key points in the approach require clarification and transparency.

We thank the reviewer for stating that the new results address the previous criticism.

1. Methods: The 'Detection of large vacuoles by microscopy' section in the Methods lacks sufficient detail for reproducibility. Please specify the ImageJ analysis workflow, including any thresholding methods, segmentation criteria for vacuoles, and any pre-processing steps applied. The method of calculating values presented in Figures 3k, 4i, and Extended Data 9b should be spelled out in detail.

We realize that our description of the method was too short, as stated by the reviewer. We have amended this point as detailed below.

The reviewer appears to assume that the analysis of vesicle size was fully automated. However, as stated in the new methods section, this was not possible with the available image analysis software. Instead, cross-sectional areas of vesicles were calculated from manually drawn regions of interests (ROIs) that follow the contours of vesicles revealed by immunocytochemistry. Cells were classified into distinct categories according to a minimum number of vesicles in a certain size range.

This is now described in detail in the new Methods section:

"CLC vacuolization assay"

HeLa WT cells (Leibniz-Institut DSMZ- Deutsche Sammlung von Mikroorganismen und Zellkulturen GmbH, Germany) or HeLa CLCN7 KO cells⁵⁸ were cultured on glass coverslips in complete medium (DMEM + 10% fetal bovine serum + 1% penicillin-streptomycin) at 37°C and 5% CO₂. Cells were transfected using JetPRIME® (Polyplus) as the transfection reagent with plasmids expressing either Venus-CLC-3a WT, or Venus-CLC-3a mutants including the transport-deficient td mutant of which the 'proton glutamate' is substituted to alanine (corresponding to E281A in the short CLC-3a isoform and E339A in the long isoform used for structural analysis), or Venus-CLC-5 WT together with WT or mutant T9A, under the control of the CMV promoter. In some experiments, T9A was substituted by CD4 as a control. 48 h after transfection, cells growing on coverslips were processed for immunocytochemistry. In experiments investigating the effect of PI(3,5)P₂, this was preceded by a 3-4 h incubation with PIKfyve

inhibitors (either 100 nM apilimod (Cat. SML2974, Sigma-Aldrich) or 1 μ M YM201636 (Cat. 371942-69-7, Merck) (both dissolved in 0.005% DMSO) in complete medium at 37 °C/5% CO₂). 0.005% DMSO in complete medium served as vehicle control in those experiments. Cells were then washed twice in cold PBS and fixed with chilled methanol for 10-15 min. Cells were extensively washed with PBS, blocked and permeabilized with 0.1% saponin in 3% sterile-filtered goat serum (NGS) (Cat. P30-1002, PAN-Biotech), 2% BSA at room temperature for 1 h. Cells were then incubated overnight at 4 °C with primary antibodies (chicken anti-GFP antibody (1: 500; Cat# GFP-1020, Aves Lab), guinea pig anti-T9A (T9AC2) (1:1000; Pineda Antibody Service, Berlin;²⁹) and mouse anti-Lamp-2 (H4B4) (1:500; Cat. Ab25631, Abcam) in 3% BSA, 0.05% saponin. Next, cells were extensively washed in 0.05% saponin-PBS, incubated with secondary antibodies coupled to different fluorophores (1: 1000; Goat anti-chicken Alexa 488 Cat. A11039, Goat anti-guinea pig Alexa 555 Cat. A21435 and goat anti-mouse Alexa 633 Cat. A21052, Invitrogen) and 1 μ g/ml DAPI at room temperature for 1 h, then washed and mounted on slides using Fluoromount-G (SouthernBiotech) and allowed to dry. Images were acquired using a LSM880 Zeiss confocal microscope using 40X water- or 63X NA 1.4 oil-immersion lenses.

Each condition (mutants or inhibitor treatments) was examined in a minimum of three biological replicates. To ensure unbiased sampling, between 8 and 16 images were randomly acquired using a 40x water-immersion objective with a Zeiss LSM880 confocal microscope for each individual experiment. To restrict the analysis to cells co-expressing both proteins, cells which displayed above-background fluorescence in both the Venus- and the T9 channel (about 20-25% of total cells) were manually selected for further analysis. Automated vesicle size determination with available ImageJ plug-ins was not possible because the software could not distinguish individual vesicles when occurring in clusters, which are frequently observed with CIC-3-enlarged vesicles. Therefore, the cross-sectional areas of individual CIC-3-containing vacuoles were calculated with ImageJ (NIH) from oval or elliptical Regions of Interest (ROIs) that were manually drawn on the perimeter (indicated by Venus-CIC-3 expression) of individual vesicles.

Enlarged vacuoles were defined by setting a minimum size threshold (cross-sectional area >1.0 μ m²). Based on the number and size of vacuoles present in each cell, we qualitatively classified cells into three distinct categories: a) cells containing small vesicles or punctae, with area < 1.0 μ m², observed in Venus-CIC-3 co-expressed with wildtype T9A; b) cells containing five or more vacuoles with cross-sectional areas between 1.0 and 1.5 μ m²; and c) cells containing five or more vacuoles with cross-sectional areas greater than 1.5 μ m². The percentage of cells falling into the different vacuole size categories was calculated and plotted as bar plot for representation. This analysis showed that enlarged vacuoles were only observed when either CIC-3 or T9A carried mutations and revealed differences between the degree of disinhibition in a semi-quantitative manner. For experiments with PIKfyve inhibitors, only cells containing ≥ 5 vacuoles with >1.5 μ m² cross-sectional area were considered as the effect appeared to be all-or-none."

Additionally, clarification is needed on whether biological or technical replicates were performed.

In addition to the Methods section pasted above, in which we describe the replicates we performed in general terms, we now state in the figure legends for each experiment how many experiments were performed, and which type of replication was performed.

For example, the legend for Figure 4 is now:

"Fig. 4| The T9A CD is necessary for inhibiting CIC-3. (a-b) Two regions of the interface between

the T9A CD and the cytosolic surface of CIC-3. Hydrogen bonds are shown as dashed lines. (c-d) Representative confocal images of CIC-3a-dependent endosomal vacuolization in cells co-expressing Venus-CIC-3a and T9A (c) or Venus-CIC-3a and T9A^{W164A} (d). Venus-CIC-3a in green, T9A in red and DAPI in blue. Zoomed view of boxed region shown at top right for Venus-CIC-3a and bottom right for T9A. Scale bars: 10 μm , and 5 μm for enlarged areas. (e) Cumulative percentage of cells displaying *punctae* ($<1 \mu\text{m}^2$), ≥ 5 moderate size vacuoles ($1-1.5 \mu\text{m}^2$) per cell, or ≥ 5 large vacuoles ($>1.5 \mu\text{m}^2$) per cell. Vacuolization was investigated in fixed immunostained cells co-expressing Venus-CIC-3a with T9A, Venus-CIC-3a with T9A^{W164A}, Venus-CIC-3a with T9A^{R172A}, and Venus-CIC-3a^{W260A} with T9A. Data are shown as mean \pm SD from 3 independent experiments (biological replicates). Source Data are provided with this paper.”

2. Figure labels and legends: The y-axis labels in Figure 3k and Extended Data Figure 9b are ambiguous and confusing. It appears to be a distribution of cells based on vacuole area in those cells. But it is unclear how the binning is done to get the distribution here. Is the ‘vacuole area’ total area occupied by all vacuoles in the cell? Or of a single ‘typical’ vacuole in the cell? ‘Typical’ assumes uniform vacuole distribution size in a cell – is that assumption justified? The y-axis label should be revised and given a precise and complete definition in the legend.

We agree that our description might have been confusing. We did not measure the total area taken up by vacuoles in any given cells, but measured cross-sectional areas of individual vesicles that were highlighted by manually drawn ROIs, and then qualitatively classified cells based on the size and number of vacuoles, using 5 vacuoles of a particular size within a cell as a threshold. This is detailed in the new Methods section pasted above. We present the exact number of cells in each class in the Source Data for the corresponding Figure.

We have revised the y-axis of Figure 4e (previously Figure 3k) and Extended Data Figure 8f (previously Extended Data Figure 9b) to “cumulative % of cells in vacuole size classes” to better describe our analyses.

To aid the reader, we have also revised the figure legends. For example, the legend for Extended Data Figure 8f is:

“(f) Cumulative percentage of cells displaying *punctae* ($<1 \mu\text{m}^2$), ≥ 5 moderate size (cross section area between $1-1.5 \mu\text{m}^2$) vacuoles per cell or ≥ 5 large ($>1.5 \mu\text{m}^2$) vacuoles per cell. Vacuolization was investigated in fixed immunostained cells co-expressing Venus-CIC-3a with T9A, Venus-CIC-3a with T9A^{R163A}, Venus-CIC-3a with T9A^{R163L}, Venus-CIC-3a^{R254A} with T9A, and Venus-CIC-3a^{K310A} with T9A, as a semiquantitative measure of CIC-3 transport inhibition by T9A or its mutants. Data are shown as mean \pm SD with 3 independent experiments each (biological replicates). Numerical data are given in Source Data provided with this paper.”

3. Include summary tables: It is difficult to read the data and error bars in the plots. While the plots provide a nice qualitative summary of the data (once points #1 and 2 are addressed), a summary table showing the values for the data presented in Figures 3k, 4i and Extended Data Figure 9b is essential for transparency and data interpretation.

Summary tables are included as Source Data for each figure.

Reviewer #2 (Remarks to the Author):

In the revised manuscript, the authors have sufficiently addressed all my concerns.

We thank the reviewer for their positive assessment of our work.

Reviewer #3 (Remarks to the Author):

The authors have addressed all comments in a satisfactory manner. The manuscript is improved and ready for publication.

We thank the reviewer for their positive assessment of our work.

Reviewer #1 (Remarks to the Author):

The authors have effectively responded to the critique and made the necessary revisions to the manuscript.

We thank the reviewer for their thoughtful criticisms and the aid in improving our manuscript.

There is just one minor typo to be fixed:

> In Figure 3e, the units should read 1.0–1.5 μm^2 and 1.5 μm^2 — the superscript 2 is currently missing.

We have added the missing superscript 2.